# SRC and TKS5 mediated podosome formation in fibroblasts promotes extracellular matrix invasion and pulmonary fibrosis

Ilianna Barbayianni[1,7], Paraskevi Kanellopoulou[1,7], Dionysios Fanidis [1], Dimitris Nastos[1], Eleftheria-Dimitra Ntouskou[1], Apostolos Galaris[1], Vaggelis Harokopos[1], Pantelis Hatzis [1], Eliza Tsitoura[2], Robert Homer [3], Naftali Kaminski [4], Katerina M. Antoniou[2], Bruno Crestani[5], Argyrios Tzouvelekis[6] & Vassilis Aidinis [1] ✉

The activation and accumulation of lung fibroblasts resulting in aberrant deposition of extracellular matrix components, is a pathogenic hallmark of Idiopathic Pulmonary Fibrosis, a lethal and incurable disease. In this report, increased expression of TKS5, a scaffold protein essential for the formation of podosomes, was detected in the lung tissue of Idiopathic Pulmonary Fibrosis patients and bleomycin-treated mice. The profibrotic milieu is found to induce *TKS5* expression and the formation of prominent podosome rosettes in lung fibroblasts, that are retained ex vivo, culminating in increased extracellular matrix invasion. *Tks5*[+/-] mice are found resistant to bleomycin-induced pulmonary fibrosis, largely attributed to diminished podosome formation in fibroblasts and decreased extracellular matrix invasion. As computationally predicted, inhibition of src kinase is shown to potently attenuate podosome formation in lung fibroblasts and extracellular matrix invasion, and bleomycin-induced pulmonary fibrosis, suggesting pharmacological targeting of podosomes as a very promising therapeutic option in pulmonary fibrosis.

Tissue fibrosis is a pathogenic process that affects most organs and constitutes a complication of many chronic diseases including cancer; such fibroproliferative disorders account for >45% of all disease-related deaths worldwide[1]. Among them, Idiopathic pulmonary fibrosis (IPF) is a chronic, progressive, interstitial lung disease affecting mostly older adults. IPF patients exhibit progressive worsening of respiratory functions, which lead to dyspnea and eventually to respiratory failure.

Histologically, IPF is characterized by lung parenchymal scarring, as evident by a usual interstitial pneumonia (UIP) profile, characterized by patchy dense fibrosis with architectural distortion and a subpleural and paraseptal preference, and is distinguished by the presence of fibroblast foci[2]. Although the etiopathogenesis of IPF remains largely elusive, the prevailing hypothesis suggests that the mechanisms driving IPF involve age-related aberrant recapitulation of developmental

[1]Institute for Fundamental Biomedical Research, Biomedical Sciences Research Center Alexander Fleming, Athens, Greece. [2]Department of Respiratory Medicine, School of Medicine, University of Crete, Heraklion, Greece. [3]Department of Pathology, Yale School of Medicine, New Haven, CT, USA. [4]Department of Internal Medicine, Yale School of Medicine, New Haven, CT, USA. [5]Department of Pulmonology, Bichat-Claude Bernard Hospital, Paris, France. [6]Department of Respiratory Medicine, School of Medicine, University of Patras, Patras, Greece. [7]These authors contributed equally: Ilianna Barbayianni, Paraskevi Kanellopoulou. ✉e-mail: V.Aidinis@Fleming.gr

programs and reflect abnormal, deregulated wound healing in response to persistent alveolar epithelial damage, resulting in the accumulation of lung fibroblasts (LFs)[3].

LFs are the main effector cells in pulmonary fibrosis, secreting exuberant amounts of extracellular matrix (ECM) components, such as different types of collagens. LFs also secrete a variety of ECM remodeling enzymes, such as matrix metalloproteinases (MMPs), thus coordinating the overall ECM structural organization and consequently the mechanical properties of the lung[4]. LF activation upon fibrogenic cues, such as TGFβ or other growth factors (including PDGF and VEGF), is characterized by the expression of alpha smooth muscle actin (αSMA/ACTA2), and/or increased collagen expression, as exemplified by COL1A1[5,6]. ECM fibrotic remodeling and resulting mechanical cues are considered as crucial stimulating and perpetuating factors for LF activation[6,7], while the chemoattraction of LFs to various signals and their resistance to apoptosis has been suggested to promote respectively their recruitment and accumulation[5,6].

Fibroblast accumulation in pulmonary fibrosis has also been suggested to be mediated by their ability to invade the underlying ECM, and increased ECM invasion of fibroblasts isolated from the lung tissue of IPF patients or animal models has been reported[8–11]. Activation of invasion, critical for embryonic development, is among the well-established hallmarks of cancer[12], and one of the many shared hallmarks between cancer cells and activated LFs[13]. Invasion critically relies on the proteolysis of the underlying ECM via invadopodia in cancer cells and podosomes in other cell types[14,15].

Podosomes are comprised of a filamentous (F)-actin-rich core enriched in actin-regulating proteins, such as the Arp2/3 complex and cortactin (CTTN), and are surrounded by a ring of scaffold proteins, most notably SH3 and PX domains 2A (SH3PXD2A; commonly known as tyrosine kinase substrate with 5 SH3 domains, TKS5)[14–16]. The effector molecules of the podosomes are various proteases, such as matrix metalloproteinases (MMPs, mainly 2, 9, and 14) that digest the ECM locally, thus stimulating the invasion and migration of podosome bearing cells[16,17].

*Tks5* expression is necessary for neural crest cell migration during embryonic development in zebrafish[18], and homozygous disruption of *Tks5* in mice resulted in neonatal death[19]. Beyond embryonic development, which heavily relies on migration and invasion, increased TKS5 expression has been reported in different types of cancers[14–16], including lung adenocarcinoma, where it was suggested to mediate metastatic invasion[20]. Pulmonary fibrosis confers one of the highest risks for lung cancer development, while many similarities between activated LFs and cancer cells have been suggested, including ECM invasion[13]. Therefore, in this report we investigated a possible role of TKS5 and podosomes in the pathogenesis of pulmonary fibrosis employing in silico analysis of publicly available human and mouse transcriptomic datasets, de novo analysis of human samples and associated clinical data, disease modeling in mice, ex vivo/in vitro human/mouse cell cultures and dedicated functional assays, as well as pharmacologic validation experiments. In this context, increased Tks5 expression is detected in both human and mouse fibrotic lungs, primarily expressed in LFs. The profibrotic milieu, is shown to induce TKS5 expression and the formation of prominent podosome rosettes in fibroblasts, culminating in increased ECM invasion. Haploinsufficient *Tks5*[+/-] mice are found resistant to BLM-induced pulmonary fibrosis, largely attributable to diminished podosome formation in LFs and decreased ECM invasion. Expression profiling reveals an ECM-podosome cross talk, and pharmacologic connectivity map analysis suggests several inhibitors that could prevent podosome formation and thus pulmonary fibrosis. Among them, inhibition of src kinase is shown to potently attenuate podosome formation in LFs, ECM invasion, as well as BLM-induced pulmonary fibrosis.

## Results

### Increased *TKS5* expression in pulmonary fibrosis

Increased *TKS5* mRNA levels were detected in silico in the lung tissue of IPF patients as compared with control samples (Fig. 1a), in most publicly available IPF transcriptomic datasets (Supplementary Table 1) at Fibromine[21], including three of the largest ones (Fig. 1b and Supplementary Fig. 1a, c). Importantly, *TKS5* mRNA expression in fibrotic lungs correlated with the expression of *COL1A1* (Fig. 1c and Supplementary Fig. 1b, d), a well-established marker of fibrotic gene expression. Confirming the in silico results, increased *TKS5* mRNA levels were detected with quantitative RNA RT-PCR (Q-RT-PCR) in lung tissue isolated from IPF patients (n = 20), as compared with COPD patients (n = 19) and healthy lung tissue (n = 9) (Supplementary Table 2 and Fig. 1d). Moreover, positive TKS5 immunostaining was detected in the lungs of IPF/UIP patients (n = 3), as opposed to control samples (n = 3), mainly localized in the alveolar epithelium and fibrotic areas (Fig. 1e and Supplementary Fig. 2). Similar conclusions were derived from the analysis of a publicly available single cell RNA sequencing (scRNAseq) dataset of lung tissue from transplant recipients with pulmonary fibrosis (n = 4) and healthy lung tissue from transplant donors (n = 8)[22]. *TKS5* mRNA expression was mostly detected in subsets of epithelial cells, basal cells and especially fibroblasts (Supplementary Fig. 1e, f), where TKS5-expressing LFs were found to belong to a *COL1A1*-expressing subpopulation (Supplementary Fig. 1g, f).

Increased *Tks5* mRNA levels, correlating with *Col1a1* mRNA levels, were also detected in the lung tissue of mice post bleomycin (BLM) administration (Fig. 1g, h), a widely used animal model of pulmonary fibrosis;[23–25] immunostaining localized Tks5 in the alveolar epithelium and fibrotic areas (Fig. 1i), as in human patients. Moreover, double immunostaining for aSMA or Col1a1, prominent activation markers of fibroblasts in both mice and humans, indicated that Tks5 localized mainly to a Col1a1 expressing fibroblast subset, as Tks5 staining overlapped with 20% of Col1a1 staining, as opposed to a 2% with aSMA staining (Fig. 1i and Supplementary Fig. 3).

Therefore, pulmonary fibrosis in both humans and mice is associated with increased *TKS5* expression, consistently correlated with the expression of *Col1a1*, especially in LFs.

### TGFβ-induced podosome rosettes is an inherent property of fibrotic LFs

TGFβ, among the main pro-fibrotic factors driving disease development in vivo, was found to stimulate *TKS5* mRNA expression in different primary normal human lung fibroblast (NHLF) clones (Fig. 2a), correlating with *COL1A1* mRNA expression (Fig. 2b); identical results were obtained from an independently derived NHLF cell line at a different lab/setting (Supplementary Fig. 4a, b), as well as from the human fibroblastic cell line MRC5 (Supplementary Fig. S4c, d). In agreement with the essential role of TKS5 on podosome formation (colocalization of F-actin with TKS5 or CTTN)[26], TGFβ, playing a prominent role in proliferation and migration of fibroblasts (Supplementary Fig. 4e–g respectively), potently stimulated the formation of podosomes in NHLFs in vitro, organized in distinctive rosettes (Fig. 2c–f and Supplementary Fig. 5a–f). Moreover, TGFβ-induced podosomes in LFs were enriched in MMP9 (Fig. 2g, h and Supplementary Fig. 5g, h), likely contributing to the increased degradation of a fluorescein-conjugated gelatin substrate (Fig. 2i, j), a nominal podosome property.

To examine if the pro-fibrotic milieu in the lungs of IPF patients, which includes TGFβ, also stimulate podosome formation in vivo, HLFs from IPF patients (Supplementary Table 3) were cultured in the absence of any stimulation and were stained for podosomes in comparison, under the same conditions (and 7–8 passages), with different NHLF lines derived from healthy tissue. Remarkably, IPF HLFs, irrespectively of cell density (Supplementary Fig. 6a), presented with

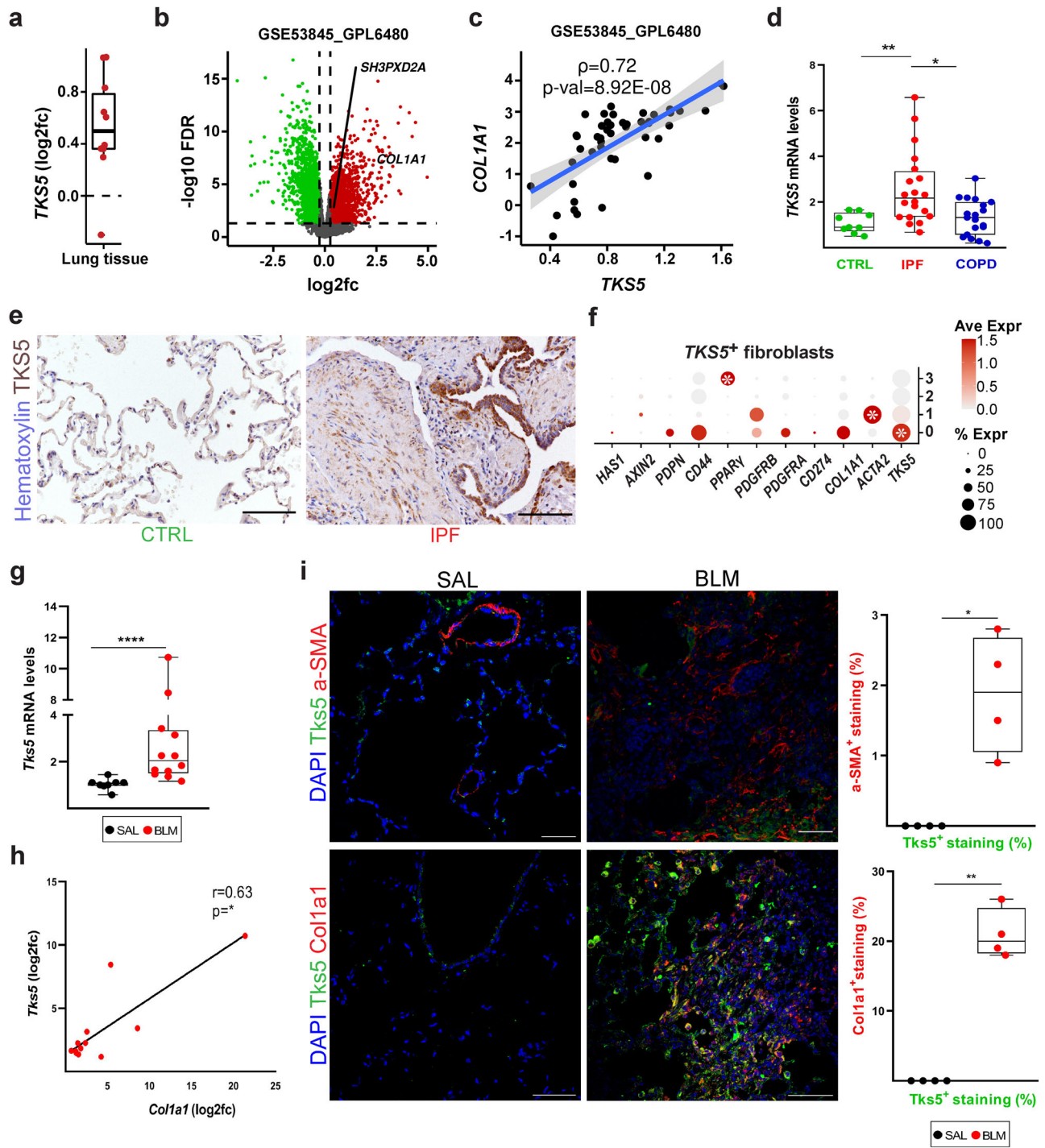

prominent podosome rosettes (Fig. 3a–d, Supplementary Fig. 6a, b, and Supplementary Movie 1), identical in structure as those stimulated in vitro by TGFβ, that persist upon prolonged culture ex vivo. As shown for TGFβ-stimulated NHLFs, IPF HLFs degraded more potently than NHLFs a fluorescein-conjugated gelatin substrate (Fig. 3e, f and Supplementary Fig. 6c).

Phenocopying the human experiments, exposure to TGFβ of primary, normal mouse lung fibroblasts (NMLFs) stimulated *Tks5* mRNA expression (Supplementary Fig. 7a), correlating with *Col1a1* expression (Supplementary Fig. 7b), the formation of podosome rosettes (Supplementary Fig. 7c, d) and the degradation of a fluorescein-conjugated gelatin substrate (Supplementary Fig. 7e, f); similar results were obtained with 3T3 embryonic fibroblasts (Supplementary Fig. 7g–k).

Moreover, and as in the case of IPF LFs, mouse primary LFs isolated post-BLM administration presented with increased *Tks5*, *Col1a1* and *Mmp9* expression (Supplementary Fig. 7l–n respectively), exhibiting prominent podosome rosettes in the absence of any stimulation (Supplementary Fig. 7o–p).

Therefore, the pro-fibrotic milieu in the lungs of IPF patients and BLM-treated mice, as well as TGFβ, induce TKS5 expression and the formation of podosome rosettes, an inherent fibrotic LF property.

## Creation of a series of obligatory and conditional knock out mice for *Tks5*

To enable functional studies on the likely role of Tks5 in pulmonary fibrosis and pathophysiology in mice, we then created a series of

**Fig. 1 | Increased *TKS5* expression in pulmonary fibrosis. a** *TKS5* mRNA expression in lung tissue from Idiopathic Pulmonary Fibrosis (IPF) patients as compared (log2FC) to controls (CTRL) in different publicly available datasets ($n = 9$) (Supplementary Table 1) at Fibromine. **b** Volcano plot from a representative large dataset (FC > 1.2, FDR < 0.05). **c** Scatter plot of *TKS5* and *COL1A1* expression in the same dataset with a fitted linear model and 95% CI; correlation was assessed with two-tailed Spearman's test ($\rho > 0.6$; $p = 8.92E{-}08$). **d** Increased *TKS5* mRNA levels in the lung tissue of IPF (Usual Interstitial Pneumonitis; UIP) patients ($n = 20$) were detected with Q-RT-PCR ($r^2 = 0.98$, $E = 97\%$), as compared with lung tissue from COPD patients ($n = 19$) and control (CTRL) lung tissue isolated from lung cancer patients ($n = 9$) (Supplementary Table 2). Values were normalized to the expression values of the housekeeping gene *B2M* and presented as fold change to CTRL values. Statistical significance was assessed with two-tailed Kruskal-Wallis test (**$p = 0.0076$, *$p = 0.0129$). **e** Increased TKS5 immunostaining in fibrotic lungs. Representative images from immunohistochemistry for TKS5 (brown) in IPF and CTRL lung tissue ($n = 3$; Supplementary Fig. 2); scale bars = 50 μm. **f** *TKS5* is expressed mainly by the *COL1A1*-expressing cluster/LF subpopulation. in a publicly available scRNAseq dataset (Reyfman, Walter et al. 2019). Statistical significance was assessed with Wilcoxon Rank Sum test (*FC > 1.2, Bonferroni corrected $p = 8.9E{-}12$ / 1.1E-10 / 2.1E-3 from left to right). **g, h** *Tks5* and *Col1a1* mRNA expression was interrogated with Q-RT-PCR ($r^2 = 0.89/0.93$; $E = 103\%/96\%$); cumulative result from 3 different experiments. Values were normalized over the expression of *B2m* and presented as fold change (log2) over control ($n = 8/12$). Statistical significance was assessed with two-tailed Mann Witney test (***$p < 0.0001$). **h** Two tailed spearman correlation plot of *Col1a1* expression in the same samples (*$p = 0.0323$; $r = 0.63$). **i.** Double immunostaining against Tks5 (green) and aSMA (Acta2) or Col1a1 (red); representative images are shown, followed by their respective quantification ($n = 4$) with Image J; scale bars=50 μm; a representative experiment out of 3 successful independent ones are shown. Statistical significance was assessed with two-tailed Welch's test (*$p = 0.0211$, **$p = 0.0013$). In all panels all samples are biologically independent; boxplots visualize the median of each distribution; upper/lower hinges represent 1st/3rd quartiles; whiskers extend no further than 1.5 * IQR from the respective hinge. Source data for all panels are provided as a Source Data file.

obligatory and conditional knock out mice for *Tks5* (*Sh3pxd2a*). The *Sh3pxd2a* locus has been already targeted by the European Conditional Mouse Mutagenesis Program (EUCOMM), aiming to knock out all mouse genes in a high throughput approach[27]. In this context, the exon 11 of the *Sh3pxd2a* gene was loxP-flanked, while a LacZ/neomycin reporter/selection cassette was placed upstream, including two FRT sites; this allele is referred to as "targeted mutation 1a" (tm1a; Supplementary Fig. 8a)[27]. The targeted ES cells were then microinjected into C57Bl/6 N blastocysts by the Welcome Trust Sanger Institute (WTSI), that were transferred in pseudopregnant females to yield the *Sh3pxd2a*tm1a(EUCOMM)Wtsi/+ heterozygous mice (Supplementary Fig. 8a). Frozen sperm of these mice was obtained from WTSI, via the INFAFRONTIER [https://www.infrafrontier.eu/] consortium[28,29] and the European Mutant Mouse Archive- EMMA [https://www.infrafrontier. eu/emma/emma-services/?keyword=sh3pxd2a&category=strains]), that was directly injected to mice in the transgenic facility of "BSRC Fleming" via IVF technology to yield the *Sh3pxd2a*tm1a(EUCOMM)WtsiFlmg/+ heterozygous mice. Mice were genotyped following the corresponding strategy from EUCOMM, that queries three different genomic fragments (lacz, WT allele, tm1a allele) by performing three independent PCR reactions (Supplementary Fig. 8b, c). Moreover, the successful targeting was also verified with long range PCR for both the 5' and 3' arms flanking the floxed region with primers against inserted sequences (Supplementary Fig. 8d, e).

To obtain the tm1b reporter allele (Supplementary Fig. 8a) *Sh3pxd2a*tm1a/Fleming/+ mice were mated with transgenic mice expressing the Cre recombinase under the control of the Cytomegalovirus (CMV) promoter in all mouse tissues and cells (*Tg-CMV-Cre*)[30]. Genetic recombination of the obtained *Sh3pxd2a*tm1b (EUCOMM)WtsiFlmg/+ mice was verified with genomic PCR (Fig. S7B, C). Q-RT-PCR in lung tissues indicated a 50% reduction of *Sh3pxd2a* mRNA levels indicating proper gene targeting (Supplementary Fig. 8g). X-gal staining, detecting LacZ expression from the promoter of Tks5 (Supplementary Fig. 8a, Tm1b), localized transcriptional *Tks5* activation (throughout development, neonatal and adult life) mainly in arterial endothelium of the lung (Supplementary Fig. 8i). No obvious gross macroscopic abnormalities were observed, while heterozygous mice were healthy and fertile.

The haploinsufficient *Sh3pxd2a*tm1b(EUCOMM)WtsiFlmg/+ and *Sh3pxd2a*tm1d/(EUCOMM)WtsiFlmg/+ (*Tks5*+/-) mice, presented with a 50% reduction of *Tks5* mRNA levels in the lung (Supplementary Fig. 8g-h), while X-gal staining (in the reporter tm1b strain) localized transcriptional *Tks5* activation (throughout development, neonatal and adult life) mainly in endothelial and smooth muscle cells in healthy conditions (Supplementary Fig. 8i). No obvious gross macroscopic abnormalities were observed, while heterozygous mice were healthy and fertile.

Intercrossing of heterozygous mice *Tks5*+/- mice yielded no homozygous knockout mice, indicating that *Tks5* has an essential role in mouse development, as previously reported for an obligatory knock out strain[19].

A similar genetic strain, *Sh3pxd2a*tm1b/(EUCOMM)Wtsi/+ was created by WTSI from the *Sh3pxd2a*tm1a(EUCOMM)Wtsi/+ mice via a cell permeable HTN-Cre[31]. *Sh3pxd2a*tm1b/(EUCOMM)Wtsi/+ heterozygous mice were systematically phenotyped from the INFAFRONTIER consortium[28,29] on our behalf, following a relative competitive call. Results indicated that *Sh3pxd2a*tm1b/(EUCOMM)Wtsi/+ mice present with no major pathophysiological abnormalities, apart from an increase of serum alkaline phosphatase.in.females.(measurements.chart[https://www.mousephenotype.org/data/genes/MGI:1298393#phenotypes-section]). Moreover, a viability primary screen phenotypic assay was performed.on.the.novel.mutant.strain.by.WTSI.(datalchart[https://www.mousephenotype.org/data/charts?accession=MGI:1298393&allele_accession_id=MGI:5636944&zygosity=homozygote¶meter_stable_id=IMPC_VIA_001_001&pipeline_stable_id=MGP_001&procedure_stable_id=IMPC_VIA_001¶meter_stable_id=IMPC_VIA_001_001&phenotyping_center=WTSI]]), confirming the requirement of *Tks5* for embryonic development.

Moreover, and to generate the conditional tm1c allele (Fig. S7A), *Sh3pxd2a*tm1a(EUCOMM)WtsiFlmg /+ mice were crossed with transgenic mice expressing the Flp recombinase under the control of the Cytomegalovirus (CMV) promoter in all mouse tissues and cells (*Tg-CMV-Flp*)[32]. Genetic recombination of the obtained *Sh3pxd2a*tm1c(EUCOMM)WtsiFlmg/+ mice was verified with genomic PCR (Supplementary Fig. 8c). To generate the conditional tm1d allele (Supplementary Fig. 8a), *Sh3pxd2a*tm1c/(EUCOMM)WtsiFlmg/ + mice were crossed with transgenic mice expressing the Cre recombinase under the control of the Cytomegalovirus (CMV) promoter in all mouse tissues and cells (*Tg-CMV-Cre*)[30]. Genetic recombination of the obtained *Sh3pxd2a*tm1d(EUCOMM)WtsiFlmg /+ mice was verified with genomic PCR (Supplementary Fig. 8c). Q-RT-PCR in lung tissue indicated a 50% reduction of *Sh3pxd2a* mRNA levels indicating proper gene targeting (Supplementary Fig. 8h).

## *Tks5* haploinsufficiency in mice attenuates BLM-induced pulmonary fibrosis

To genetically dissect the likely role of Tks5 in pulmonary fibrosis, BLM was administered to 8-10-week-old C57Bl6/J *Tks5*+/- mice and WT littermates (Fig. 4a, b), which were sacrificed 14 days post BLM (at the peak of the disease in the local settings), as previously described[24]. No weight loss, an overall systemic health indicator, was observed in *Tks5*+/- mice (Fig. 4c), as opposed to wt mice. Vascular leak and pulmonary edema were significantly reduced in *Tks5*+/- mice, as indicated by the total protein concentration in the bronchoalveolar lavage fluid (BALF), determined with the Bradford assay (Fig. 4d). Inflammatory cells in the BALF, as measured by hematocytometer, were found

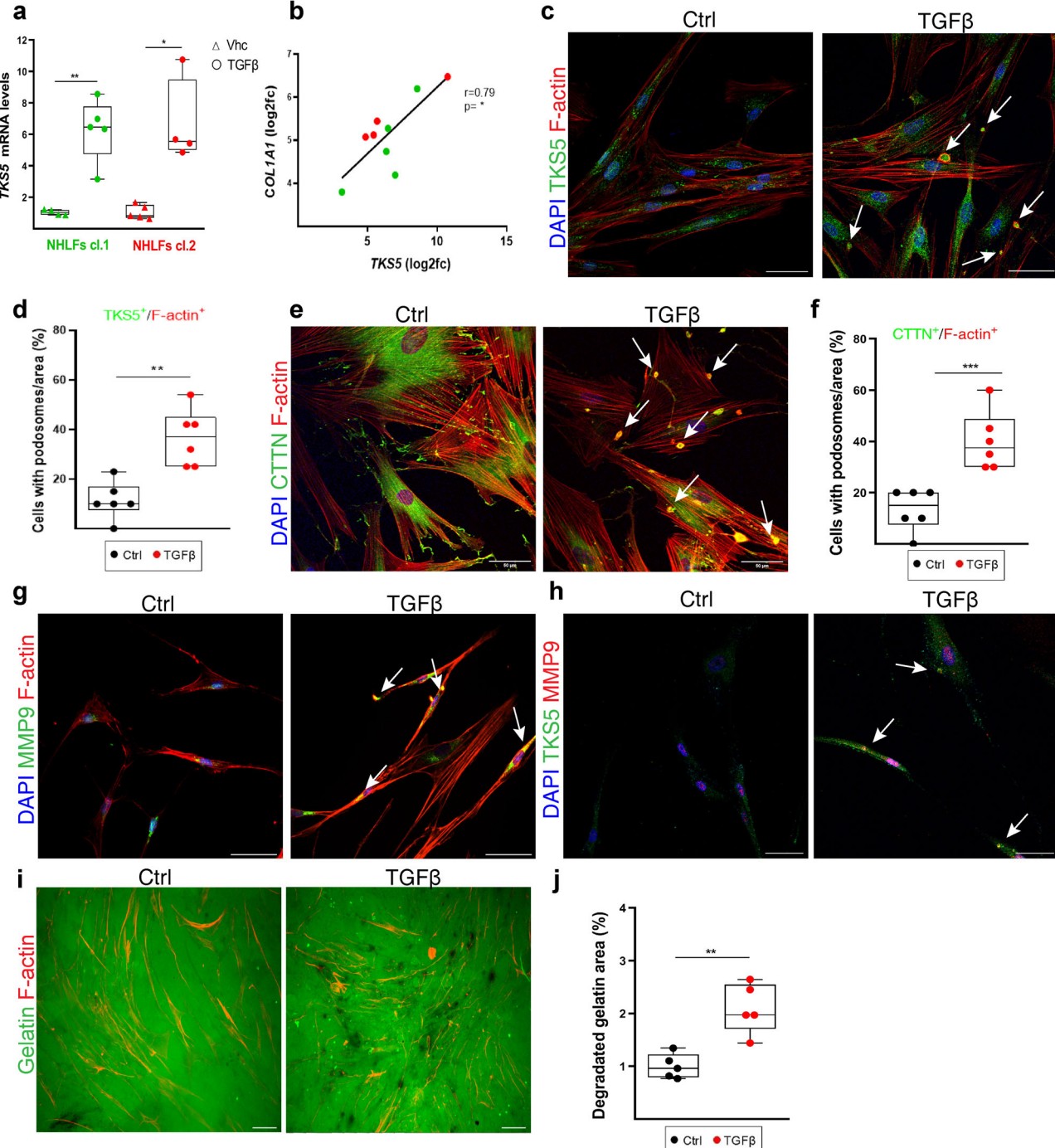

**Fig. 2 | TGFβ induces the formation of podosome rosettes in normal human lung fibroblasts (NHLFs).** Serum starved, sub-confluent (70–80%), primary NHLFs were stimulated with recombinant human TGFβ (10 ng/ml) for 24 h; a representative experiment out of 4 successful independent ones is shown. **a**, **b** *TKS5* and *COL1A1* mRNA expression was interrogated with Q-RT-PCR ($r^2$ = 0,94/0,92; E = 98,3%/93% respectively) in two NHLF clones (cl.l, cl.2). Values were normalized to the expression values of the housekeeping gene *B2M* and presented as fold change over control; $n$ = 4/5/4/4; statistical significance was assessed with two-tailed Welch's test (a/cl.1) and two-tailed Mann Whitney test (a/l.2); $p$ = 0.0012, $p$ = 0.0159 respectively. **b** Two tailed Pearson correlation plot of *COL1A1* expression in the same samples ($p$ = 0.0116; r = 0.79). **c**–**j** Representative composite images from double immunostaining, and respective quantifications, for: **c** F-actin/

TKS5 (red/green), **e** F-actin/Cortactin (CTTN; red/green), **g** F-actin/MMP9 (red/green), **h** TKS5/MMP9 (green/red). Cells are counterstained with DAPI; scale bars 50 μm; arrows indicate representative podosomes; separate images and proof of colocalization of signals can be found at Supplementary Fig. 5. **d, f**. Quantification of the number of podosome-containing cells per optical field ($n$ = 6); statistical significance was assessed with two-tailed t-test; $p$ = 0.0011, $p$ = 0.0009. **i** Representative images of the TGFβ-induced degradation (black holes) of a fluorescein-conjugated gelatin substrate. **j** Quantification of gelatin degradation area, normalized to control ($n$ = 5); statistical significance was assessed with two-tailed t-test; $p$ = 0.0016. Source data for all relative panels (a, b, d, f, j) are provided as a Source Data file.

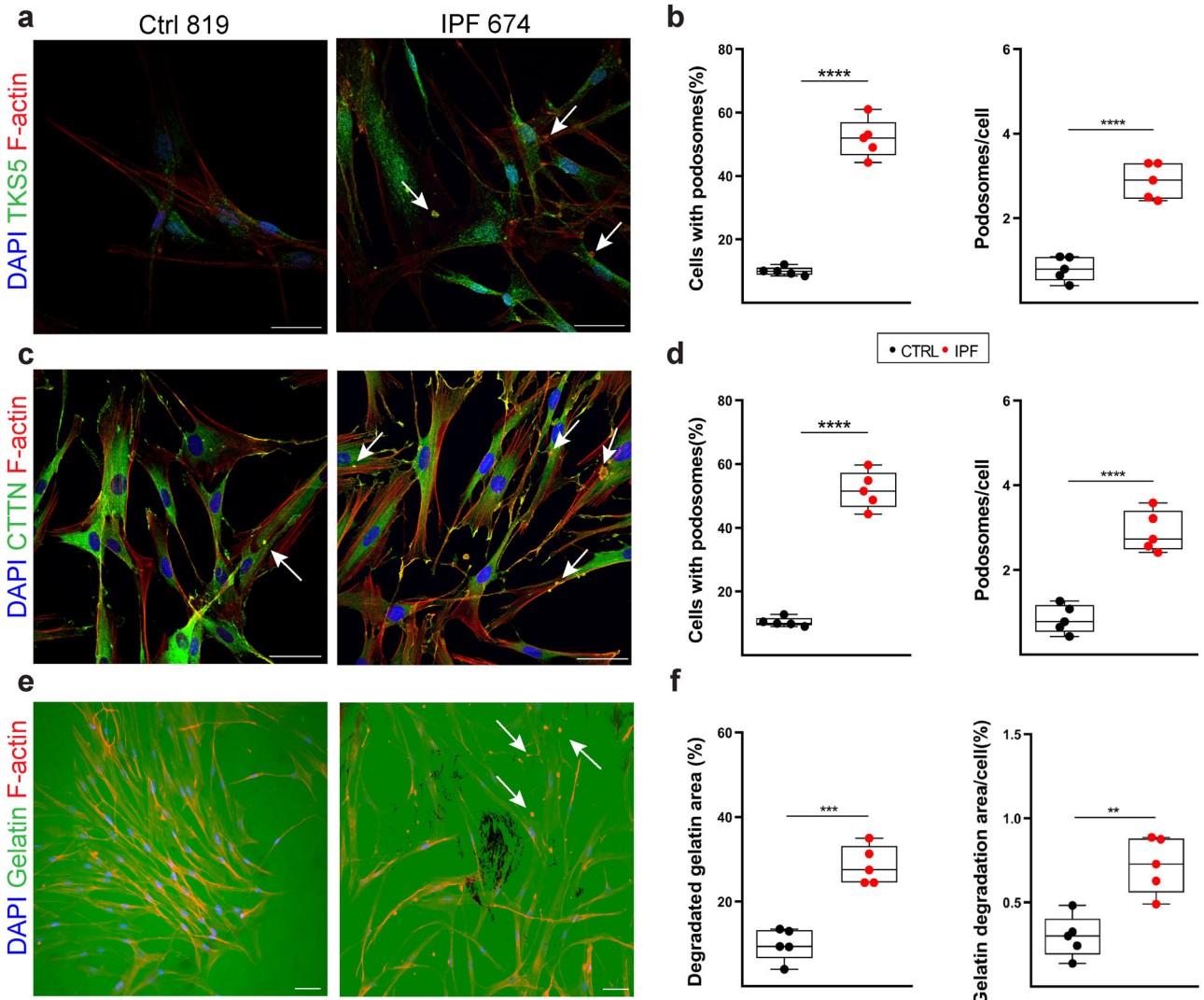

**Fig. 3 | The formation of extracellular matrix (ECM) degrading podosome rosettes is an inherent property of IPF human lung fibroblasts (HLFs).** Serum starved, sub-confluent (70-80%), primary IPF-HLFs and normal HLFs (NHLFs) were immunostained for F-actin (red) and (**a**) TKS5 (green) or (**c**) cortactin (CTTN; green) and counter stained with DAPI (blue); n = 5; scale bars = 50 μm. Representative images from representative clones are shown. **b**, **d** Cumulative quantification of the number of podosome-containing cells (%) and the number of podosomes per cell per optical field respectively. Statistical significance was assessed with two-tailed $t$-test (**b**) or two-tailed Welch's test (**d**) (****$p < 0.0001$). **e** The same clones were cultured on a fluorescein-conjugated gelatin substrate (green) and were stained for

F-actin (red) and counter stained with DAPI (blue); representative images are shown. **f** Quantification of the percentage of the degraded gelatin for all clones cumulatively, and the quantification of gelatin degradation area per cell, as quantified with ImageJ; statistical significance was assessed with two-tailed $t$-test; ***$p = 0.0001$/**$p = 0.0020$; additional clones and controls are shown at Supplementary Fig. 6. In all panels all samples are biologically independent; boxplots visualize the median of each distribution; upper/lower hinges represent 1st/3rd quartiles; whiskers extend no further than 1.5 ˙IQR from the respective hinge. Source data for all relative panels (**b**, **d**, **f**) are provided as a Source Data file.

significantly reduced in $Tks5^{+/-}$ mice (Fig. 4e); so were soluble collagen BALF levels, as determined by the Sirius red assay (Fig. 4f), in concordance with *Col1a1* mRNA expression in the lung tissue from the same mice, as determined with Q-RT-PCR (Fig. 4g, h). Histological analysis revealed decreased collagen deposition in $Tks5^{+/-}$ mice post BLM, as quantified by Sirius red/Fast green staining (Fig. 4i), and fewer peribronchiolar and parenchymal fibrotic regions were detected (Fig. 4i), as reflected in the Ashcroft score (Fig. 4j); similar conclusions were drawn upon the histological evaluation of Precision Cut Lung Slices (PCLS) prepared from the same mice and cultured ex vivo (Fig. 4i). The relative protection from the BLM-induced tissue architecture distortion upon the genetic reduction of $Tks5$ expression was also reflected in lung respiratory functions, as measured with FlexiVent (Fig. 4k–m). Therefore, Tks5 expression, and likely the formation of podosomes, were shown to

have a major role in BLM-induced pulmonary fibrosis, and therefore likely IPF.

### $Tks5$ haploinsufficiency in mouse LFs decreases ECM-regulated podosome formation and ECM invasion

To functionally dissect the relative protection of $Tks5^{+/-}$ mice from BLM-induced pulmonary fibrosis, primary LFs were isolated from littermate wt and $Tks5^{+/-}$ mice and were exposed to TGFβ, as before. $Tks5^{+/-}$ LFs, expressing ~50% of Tks5 (Fig. 5a), presented with decreased numbers of podosomes in response to TGFβ (Fig. 5b, c), reaffirming the seminal role of Tks5 in podosome formation[26], as well as with decreased proliferation (24 h; Fig. 5d). As podosomes are known to promote ECM invasion, we then examined the ability of LFs to invade acellular ECM (aECM) prepared from the lungs of mice (Supplementary Fig. 9a, b), in a transwell invasion chamber (6 h; Fig. 5e). The reduction of podosomes

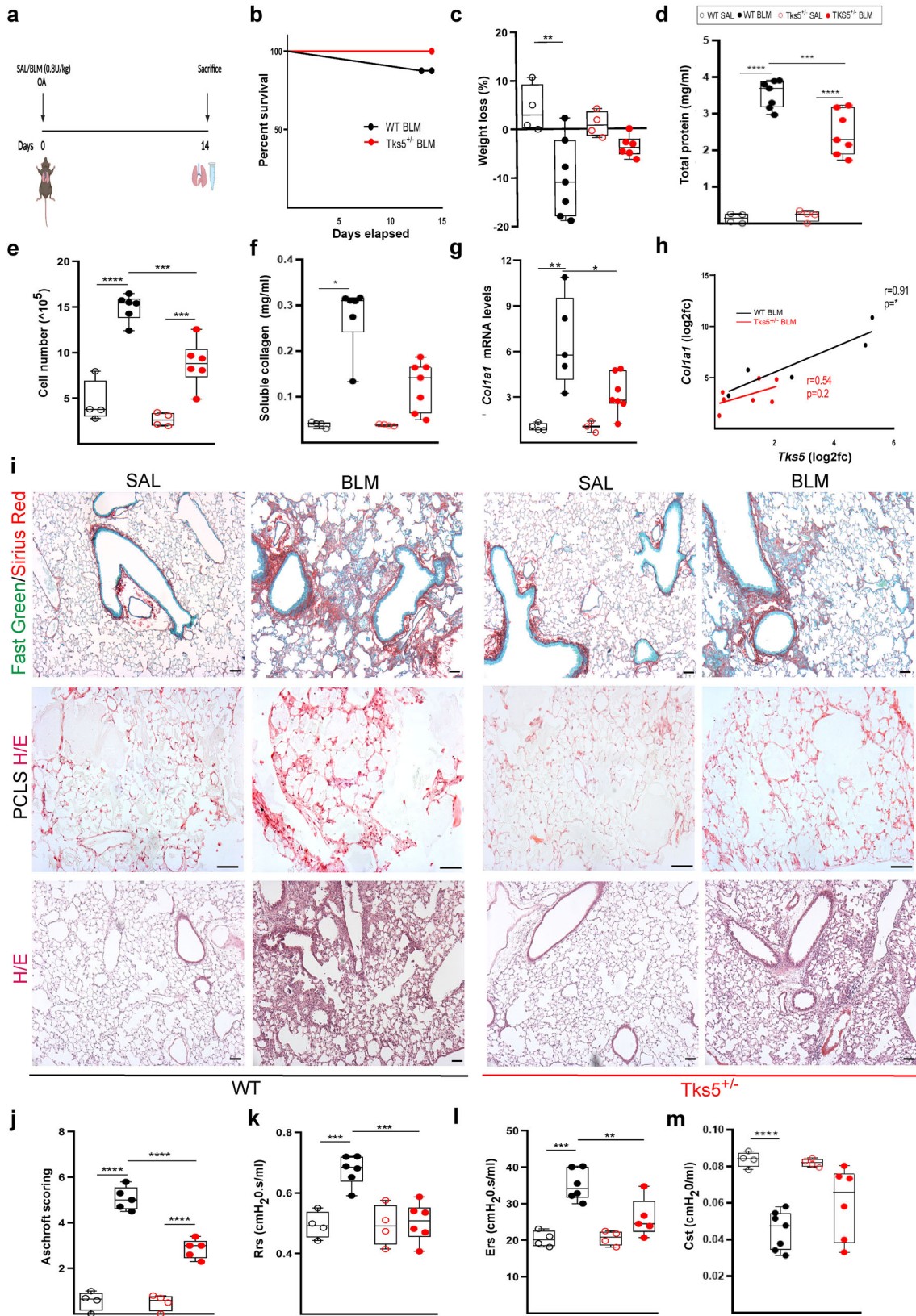

was associated with a decreased TGFβ-induced invasion of *Tks5*[+/-] LFs in aECM (Fig. 5f). Moreover, reaffirming in mice the inherent character of podosome formation in LFs, post BLM *Tks5*[+/-] LFs presented with reduced numbers of podosomes in comparison with wt LFs isolated from littermate mice (Fig. 5g, h), resulting in defective aECM invasion (Fig. 5i). Therefore, the in vivo demonstrated pathogenic role of Tks5 in

pulmonary fibrosis includes the formation of podosomes in LFs and the promotion of their ECM invasion.

To obtain additional mechanistic insights, wt and *Tks5*[+/-] LFs were exposed to TGFβ, as above, and their global expression profile was interrogated with 3' UTR RNA sequencing (Quant-Seq LEXO-GEN). Differential expression analysis between TGFβ-induced *Tks5*[+/-

**Fig. 4 | *Tks5* haploinsufficiency in mice attenuates bleomycin (BLM)-induced pulmonary fibrosis (PF). a** Schematic presentation of the BLM-induced PF model. **b**. Kaplan Meyer survival curve post BLM administration. **c**. Weight change post BLM administration; *n* = 4/7/4/6. Statistical significance was assessed with two-tailed one-way ANOVA; **p* = 0.031. **d** Total protein concentration in bronchoalveolar lavage fluids (BALFs), as determined with the Bradford assay; *n* = 4/7/4/7. Statistical significance was assessed with two-tailed one-way ANOVA; ****p* < 0.0001, ***p* = 0.0009. **e** Inflammatory cell numbers in BALFs, as counted with a hematocytometer; *n* = 4/6/4/6. Statistical significance was assessed with two-tailed one-way ANOVA; ****p* < 0.0001, ***p* = 0.0002/0.0008. **f** Soluble collagen levels in the BALFs were detected with the direct red assay; *n* = 4/6/4/7. Statistical significance was assessed with two-tailed Kruskal Wallis; **p* = 0.0124. **g**, **h** *Tks5* and *Col1a1* mRNA expression was interrogated with Q-RT-PCR; *n* = 4/5/3/7; values were normalized over the expression of the housekeeping gene *B2m* and presented as fold change over control. Statistical significance was assessed with two-tailed one-way ANOVA; ***p* = 0.0012, **p* = 0.0207. **h** Two tailed pearson correlation plot of *Col1a1* expression in the same samples; **p* = 0.0342; *r* = 0.91/0.54. **i** Representative images from lung sections of murine lungs of the indicated genotypes, stained with Fast Green/Sirius Red (green/red; first row), from Hematoxylin & Eosin (H&E)-stained Precision cut lung slices (PCLS) (second row) and H&E-stained lung sections (third row); scale bars 50 μm. **j**. Quantification of fibrosis severity in H/E stained lung sections via Ashcroft scoring; *n* = 4/5/4/5. Statistical significance was assessed with two-tailed one-way ANOVA; ****p* < 0.0001. **k** Rrs, mean respiratory system resistance as measured with Flexivent; *n* = 4/6/4/6. Statistical significance was assessed with two-tailed one-way ANOVA; ***p* = 0.0008/0.0004. **l** Ers, mean respiratory system elastance as measured with Flexivent; *n* = 4/6/4/5. Statistical significance was assessed with two-tailed one-way ANOVA; ***p* = 0.0002, **p* = 0.0081. **m** Cst, mean static lung compliance as measured with Flexivent; *n* = 4/7/4/6. Statistical significance was assessed with two-tailed one-way ANOVA followed by Welch's correction; ****p* < 0.0001. In all panels cumulative results from 2 different experiments are shown; all samples are biologically independent; boxplots visualize the median of each distribution; upper/lower hinges represent 1st/3rd quartiles; whiskers extend no further than 1.5*IQR from the respective hinge. Source data for all relative panels (**a**–**h**, **j**–**m**) are provided as a Source Data file.

and wt LFs, revealed 3648 differentially expressed genes (DEGs; FC > 1.2, FDR corr. *p* < 0.05; Supplementary Data 1 and Supplementary Fig. 10a); among them 418 DEGs have been previously associated with pulmonary fibrosis, as detected with text mining of abstract co-occurrence of identified DEGs with fibrosis keywords (Supplementary Data 1). *Stat1*, *Cebpa*, and *Ar* transcription factors (TFs), where found downregulated in *Tks5*[+/-] LFs along with several of their target genes (Supplementary Data 1 and Supplementary Fig 10b). Gene set enrichment analysis (GSEA) performed on DEGs revealed that the most affected cellular components (CC), molecular functions (MF) and biological processes (BP) all relate to the ECM (Fig. 6a and and Supplementary Data 2). "Collagen containing ECM" (GO:0062023) was most prominent due to the down regulation of several ECM related genes such as collagens and MMPs/TIMPS/Adamts (Fig. 6b and Supplementary Fig. 10c), In this context and given the observed consistent correlation of *Tks5* and *Col1a1* expression, *Tks5*[+/-] LFs post BLM, containing fewer podosomes and exhibiting defective aECM invasion (Fig. 5g–i), were found to produce significantly less Col1a1 (Fig. 6c). Vice versa, culture of primary NMLFs on Col1a1-rich aECM prepared from the lungs of mice post BLM (Supplementary Fig. 9c), stimulated *Tks5* expression (Fig. 6d, e) and the formation of podosomes (Fig. 6f), and further stimulated *Col1a1* expression (Fig. 6g), indicating an ECM-podosome cross talk in the perpetuation of LF activation.

### Src-inhibition potently reduces podosome formation and attenuate pulmonary fibrosis

To identify pharmaceutical compounds that can induce a similar transcriptional profile as that of the defective in ECM invasion *Tks5*[+/-] LFs, the TGFβ-induced *Tks5*[+/-] LFs profile was queried against the connectivity map (CMap) LINCSL1000 database (Fig. 7a), a public resource that contains >10[6] gene expression signatures of different cell types treated with a large variety of small molecule compounds[33]. Among the identified compounds with similar expression signatures, several have already been shown to have a positive effect in disease pathogenesis in animal models (Fig. 7a and Supplementary Table 4). The identified possible therapeutic targets include the PDGF and VEGF receptors, which are pharmacologically targeted by the current IPF standard of care (SOC) compound nintedanib[34]. More importantly, the list also includes an inhibitor of Src, a TGFβ/PDGF-inducible, non-receptor tyrosine kinase essential for podosome formation[35]. To verify the in silico findings in our experimental settings, TGFβ-activated NHLFs were incubated with nontoxic, increasing concentrations of nintedanib and A-419259, a commercially available src inhibitor. Both nintedanib but especially A-419259 reduced both *TKS5* and *COL1A1* expression (Fig. 7b–e), as well as podosome formation (Fig. 7f, g) and aECM LF invasion (Fig. 7h).

To examine possible therapeutic effects of src inhibition in pulmonary fibrosis, we generated mouse precision cut lung slices (PCLS) post BLM (d11) administration, which were then incubated with A-419259 for 3 consecutive days, resulting in the attenuation of pulmonary fibrosis (Fig. 7i). Moreover, the same inhibitor was administered for 6 days by inhalation (4 ml of 0182 mg/ml for 5 mins/6 mice, corresponding to 2 mg/Kg per mouse) to conscious, softly re-strained mice, in a therapeutic mode (7d post BLM; Fig. 8a); no lethality was observed (Fig. 8b); minimal changes were observed in weight loss (Fig. 8c). Remarkably, src inhibition decreased, pulmonary edema (Fig. 8d) and inflammation (Fig. 8e), and attenuated Col production (Fig. 8f, g). Accordingly, src inhibition attenuated collagen deposition in the lung tissue, and prevented BLM-induced architectural distortion (Fig. 8h, i). Therefore, the TKS5-mediated podosome formation is a druggable pathologic process, which can be potently targeted by Src inhibition.

## Discussion

Increased *TKS5* expression was detected, for the first time in a non-malignant disease[15], in the lung tissue of IPF patients and BLM-treated mice (Fig. 1 and Supplementary Figs. 1–2). Increased *TKS5* expression has been previously reported, beyond cancer cell lines, in lung adenocarcinoma[20], further extending the similarities of IPF and lung cancer[13]. *TKS5* mRNA expression in the lung tissue, of both humans and mice, correlated with the mRNA expression of *COL1A1*, a hallmark of deregulated expression in IPF, while *TKS5* expression in fibrotic lungs was predominantly localized in the alveolar epithelium and COL1A1-expressing LFs (Fig. 1 and Supplementary Figs. 1-3), pending larger scale immunohistochemical studies.

TGFβ, the prototypic pro-fibrotic factor, was found to be a very potent inducer of *TKS5* expression and podosome formation in fibroblasts (NHLFs, NMLFs, MRC5, 3T3)(Fig. 2, Supplementary Figs. 4-5), as previously reported only for THP-1 macrophages[36,37] and primary aortic endothelial cells[38,39]. Other well established pro-fibrotic growth factors in the lung have been reported to modulate podosome formation in different cell types: PDGF in synovial fibroblasts[40] and smooth muscle cells[41], and VEGF in endothelial cells[42], suggesting that they could exert similar stimulatory effects on LFs. Moreover, PGE2, which suppresses pulmonary fibrosis[43], has been reported to promote the dissolution of podosomes in dendritic cells[44], suggesting that the diminished PGE2 levels in IPF[43] also favor the formation of podosomes. Remarkably, the formation of podosomes in LFs was shown to be an inherent property of IPF and post BLM LFs that can be maintained in culture in the absence of any stimulation (Fig. 3, Supplementary Figs. 6-7). In agreement, increased invadosome formation was very recently reported in IPF LFs (3–7 passages), correlating with fibrosis severity[45]. Therefore, podosome formation is an unappreciated central

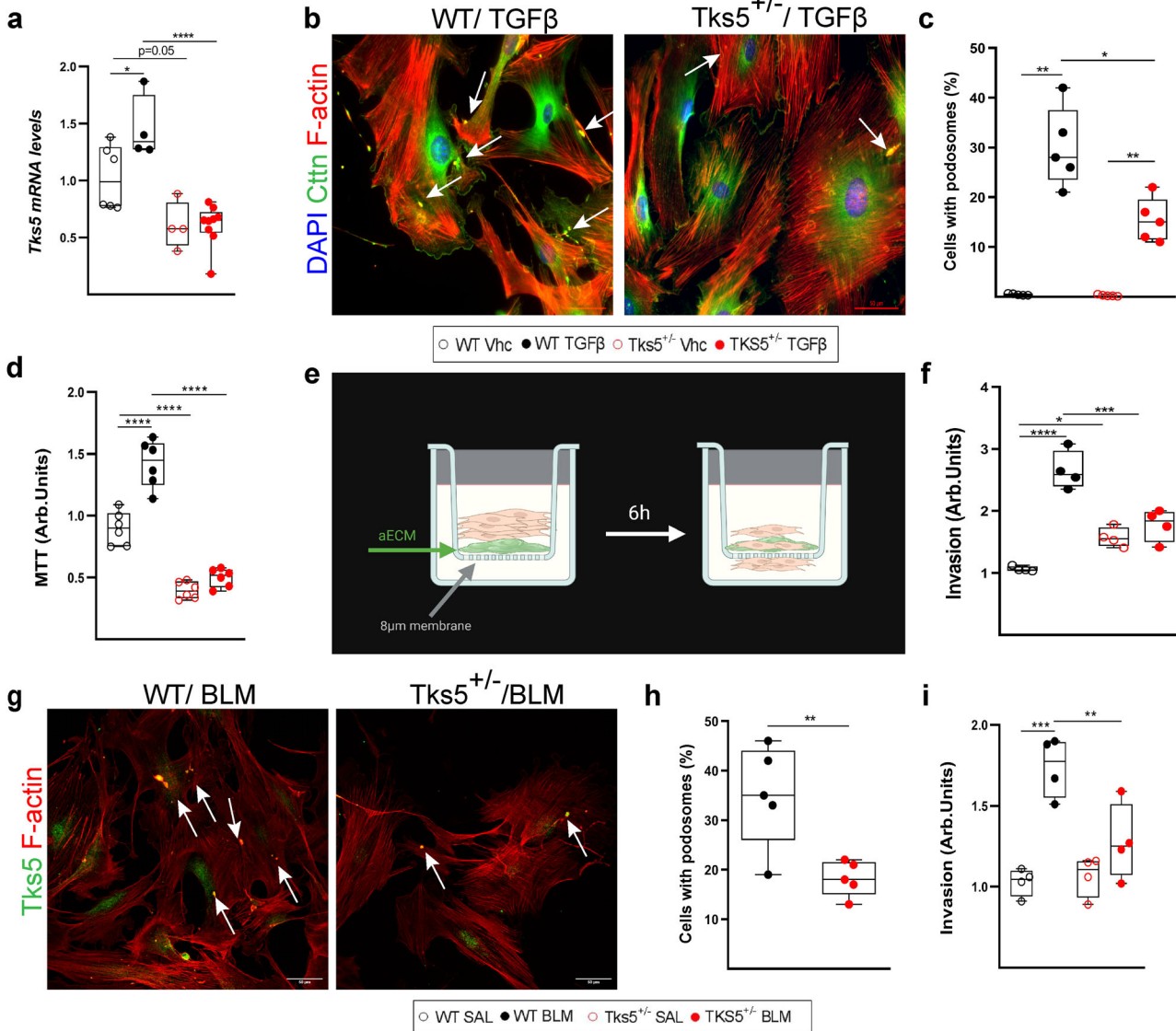

**Fig. 5 | Tks5 haploinsufficiency in mouse lung fibroblasts (LFs) decreases podosome formation and extracellular matrix (ECM) invasion.** Serum starved primary normal mouse LFs (NMLFs) from WT and Tks5+/- mice were stimulated with recombinant TGF-β1 (10 ng/ml for 24 h). **a** *Tks5* mRNA expression was interrogated with Q-RT-PCR; $n = 6/4/4/9$. Values were normalized over the expression of the housekeeping gene *B2m* and presented as fold change over control. Statistical significance was assessed with two-tailed one-way ANOVA; $^*p = 0.0464$, $^{****}p < 0.0001$. **b** Representative composite images from double immunostaining for F-actin (red) and Cortactin (Cttn; green) counter stained with DAPI (blue); arrows indicate representative podosomes. **c** Quantification of the number of podosome-containing cells per optical field; $n = 5$. Statistical significance was assessed with two-tailed one-way ANOVA followed by Welch's correction; $^*p = 0.004/0.0053$ $^*p = 0.0411$. **d** TGFβ-induced NMLFs proliferation was assessed with the MTT assay; $n = 6$. Statistical significance was assessed with two-tailed one-way ANOVA; $^{****}p < 0.0001$. **e** Schematic presentation (biorender.com) of LFs invasion into aECM, upon TGFB stimulation. After 6 h, cells that had invaded into the lower surface of

the upper chamber were stained, lysed and absorbance values were measured. **f** Invasion capacity of NMLFs, upon TGF-β stimulation ($n = 4$), as detected with the transwell invasion assay. Statistical significance was assessed with two-tailed one-way ANOVA; $^{****}p < 0.0001$, $^*p = 0.0266$, $^{***}p = 0.0005$. **g** Representative composite images from double immunostaining for F-actin (red) and Tks5 (green) in NMLFs isolated from WT and *Tks5+/-* mice, post bleomycin (BLM) administration; arrows indicate representative podosomes; scale bars 50 μm. **h** Quantification of the number of podosome-containing cells per optical field ($n = 5$). Statistical significance was assessed with two-tailed t-test; $^{**}p = 0.009$. **i** Invasion capacity of LFs post BLM, as detected with the transwell invasion assay; $n = 4$. Statistical significance was assessed with two-tailed one-way ANOVA; $^{***}p = 0.0003$ $^{**}p = 0.0099$. In all panels, all samples are biologically independent; boxplots visualize the median of each distribution; upper/lower hinges represent 1st/3rd quartiles; whiskers extend no further than 1.5 $^*$IQR from the respective hinge. Source data for all panels are provided as a Source Data file.

response of LFs to pro-fibrotic factors and a major inherent characteristic of IPF LFs, likely contributing to their accumulation and the formation of foci, a hallmark of UIP/IPF.

*Col1a1* mRNA levels were found to consistently correlate with *Tks5* mRNA levels in both humans and mouse lung tissue or LFs (Figs. 1–3 and 5, and Supplementary Figs. 1, 3, and 9). *Tks5+/-* LFs were shown to produce less Col1a1 post BLM (Fig. 6c), as also reflected in the reduced overall collagen deposition in the lungs of *Tks5+/-* mice (Fig. 4g). Culture

of LFs on Col1a1-rich aECM promoted *Tks5* expression and podosome formation (Fig. 6d−), as well as further *Col1a1* expression, emphasizing a TGFβ-induced Col1a1-podosomes interdependency in the context of the suggested crosstalk of ECM with podosomes[46]. Accordingly, Col1 has been shown to stimulate Tks5-dependent growth, while the degree of collagen fibrilization has also been reported to have a decisive effect on podosome formation[47], likely though the Discoidin domain receptors (DDRs) that mediate collagen binding[48]. The expression of both

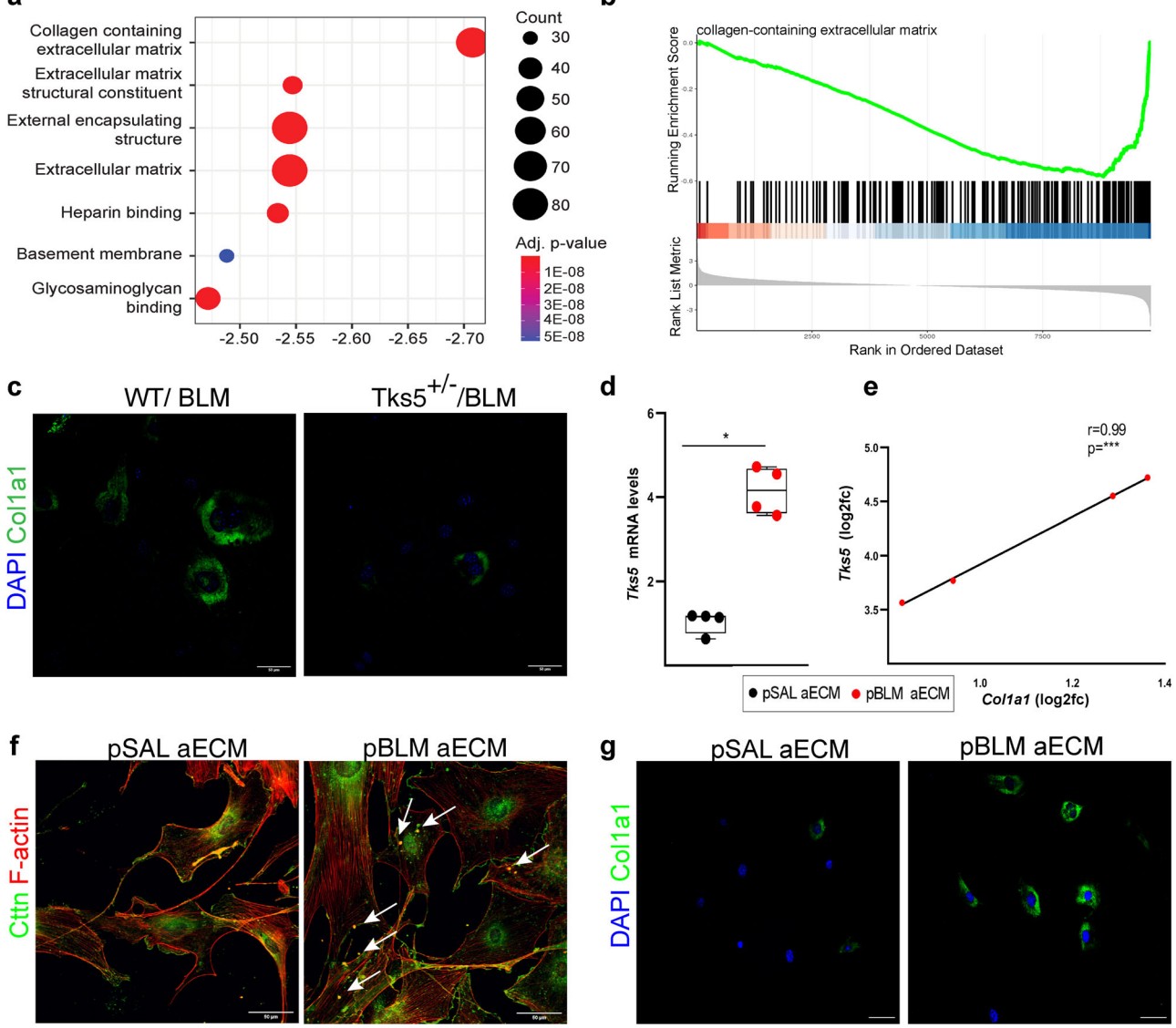

**Fig. 6 | Tks5 haploinsufficiency in mouse lung fibroblasts (LFs) disrupts extracellular matric (ECM) homeostasis, that critically controls podosome formation and ECM invasion. a** ECM-related gene ontology components are enriched for genes down-regulated in TGFβ stimulated Tks5$^{+/-}$ LFs compared to their WT TGFβ treated counterparts. Gene-set enrichment analysis (GSEA) on expression data pre-ranked according to their fold change values. **b** Collagen containing extracellular matrix is the term most enriched in down-regulated genes according to gene-set enrichment analysis (GSEA). **c** Serum starved WT and Tks5$^{+/-}$ LFs were immunostained for Col1a1 (green) and counter stained with DAPI (blue). Representative images are shown; scale bars = 50 μm. **d**, **e** Serum starved WT primary LFs were cultured in post saline (pSAL) and post bleomycin (pBLM) aECM. *Tks5* and *Col1a1* mRNA expression was interrogated with Q-RT-PCR; *n* = 4; values

were normalized over the expression of the housekeeping gene *B2m* and presented as fold change over control. Statistical significance was assessed with two-tailed Mann-Whitney test; $^{*}p$ = 0.0286. **e** Two tailed pearson correlation plot of *Col1a1* expression in the same samples; $^{***}p$ = 0.0004, *r* = 0.99. **f**, **g** Representative composite images from double immunostaining for F-actin (red) and Cortacin (Cttn; green; **f**) or Col1a1 (**g**; green) counter stained with DAPI (blue); arrows indicate representative podosomes; scale bars = 50 μm. In all panels, representative experiment out of 2 successful independent ones are shown; all samples are biologically independent; boxplots visualize the median of each distribution; upper/lower hinges represent 1st/3rd quartiles; whiskers extend no further than 1.5 *IQR from the respective hinge. Source data for all panels are provided as a Source Data file.

DDR1 and 2 were found downregulated in TGFβ-induced *Tks5*$^{+/-}$ LFs (Supplementary Data 1), suggesting that DDR signaling, and consequent tyrosine kinase activation could mediate the observed TGFβ-Col1a-induced podosome formation. While the reverse signaling from the overproduction of collagen and the regulation of DDR1/2 should be further researched in the context of IPF, the therapeutic potential of DDR inhibitors, currently explored for metastatic cancer[49], should be also investigated.

Another potent podosome inducer is tropoelastin[46], the soluble precursor of the cross-linked ECM protein elastin (Eln) whose expression was found accordingly downregulated in podosome deficient

*Tks5*$^{+/-}$ LFs (Supplementary Data 1). Moreover, and beyond individual fibrosis-modulating factors, the stiff fibrotic post-BLM aECM was shown to stimulate *Tks5* expression and podosome formation in LFs and to perpetuate the increased expression of Col1a1 (Fig. 6). In agreement, increased substrate rigidity, modeled with gelatin or polyacrylamide, has been previously shown to promote invadopodia activity[50]. Therefore, the formation of podosomes in LFs upon mechanical cues from the stiff ECM of fibrotic lungs is a major component of the suggested crosstalk of ECM with fibroblasts[51,52], especially considering the age-related increase of ECM stiffness in the lungs[7], and the suggested role of mechanosensitive signaling in LF activation and pulmonary fibrosis[53].

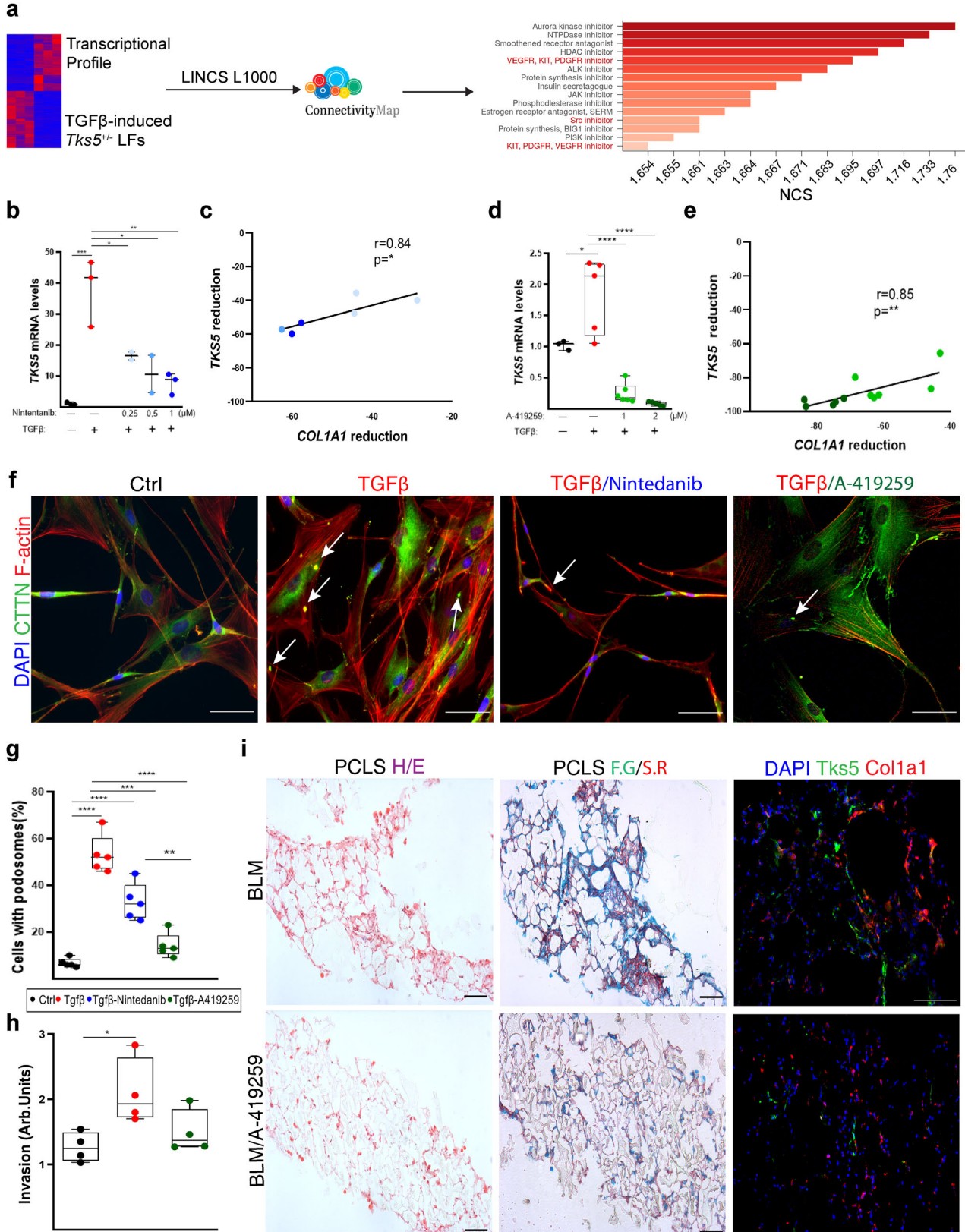

Given the necessity of TKS5 expression for embryonic development, the results presented here suggest that the formation of podosomes in LFs is a developmental program that gets aberrantly re-activated in IPF from pro-fibrotic factors, perpetuating LF activation and stimulating ECM invasion and LF accumulation. The re-activation of developmental pathways is a common theme in organ

fibrosis, e.g. wnt and ATX/LPA signaling in pulmonary fibrosis, providing increased plasticity and regeneration potential, as well as increased cell proliferation, migration and invasion.

Ubiquitous genetic Tks5 haploinsufficiency was shown to attenuate BLM-induced pulmonary fibrosis with a plethora of readout assays (Fig. 4), and *Tks5*[+/-] LFs were shown to form fewer podosomes,

**Fig. 7 | Src-inhibition potently reduces podosome formation, extracellular matrix (ECM) invasion and attenuates pulmonary fibrosis.** Serum starved, primary normal human lung fibroblasts (NHLFs) were pretreated for 1 h with A-419259 (SRC inhibitor) and Nintedanib, and then stimulated with recombinant human TGFβ (10 ng/ml) for 24 h. **a** Graphical representation of connectivity (CMap) analysis using LINCS1000 resource of the TGFβ-induced $TKS5^{+/-}$ expression profile. **b**, **c** TKS5 and COL1A1 mRNA expression was interrogated with Q-RT-PCR. Values were normalized to the expression values of the housekeeping gene B2M and presented as fold change over control; $n = 3/3/3/3$. Statistical significance was assessed with two-tailed one-way ANOVA; $^{***}p = 0.0008$, $^{*}p = 0.0397/0.0107$ $^{**}p = 0.003$. **c** Two tailed Pearson correlation plot of COL1A1 reduction in the same samples ($^{*}p = 0.0202$, $r = 0.84$). **d**, **e** TKS5 and COL1A1 mRNA expression was interrogated with Q-RT-PCR. Values were normalized over the expression values of the housekeeping gene B2M and presented as fold change over control; $n = 3/5/6/5$. Statistical significance was assessed with two-tailed one-way ANOVA; $^{*}p = 0.0202$, $^{****}p < 0.0001$. **e** Two tailed spearman correlation plot of COL1A1 reduction in the same samples; $^{**}p = 0.0018$, $r = 0.85$. **f** Representative composite images from double immunostaining for F-actin (red) and Cortactin (cttn; green) counter stained with DAPI (blue); arrows indicate representative podosomes; scale bars = 50 μm. **g** Quantification of the number of podosome-containing cells per optical field; $n = 5$. Statistical significance was assessed with two-tailed one-way ANOVA; $^{****}p < 0.0001$, $^{***}p = 0.0006$, $^{**}p = 0.0014$. **h** Invasion capacity of NHLFs ($n = 4$), upon A-419259 pretreatment and TGF-β stimulation, as detected with the transwell invasion assay; $n = 4$. Statistical significance was assessed with two-tailed one-way ANOVA; $^{*}p = 0.0294$. **i** Src-inhibition attenuates pulmonary fibrosis in mouse precision cut lung slices (PCLS) generated post BLM (d11) administration. Treatment with A-419259, was administered in the first 24 h after slicing for 3 consecutive days. Representative images from PCLS stained with H&E, Fast green/Sirius red (F.G./S./R.; green/red) and from double immunostaining for Tks5 and Col1a1 (green/red) are shown; scale bars=50 μm. In all panels, representative experiment out of 2 successful independent ones are shown. In all panels, all samples are biologically independent; boxplots visualize the median of each distribution; upper/lower hinges represent $1^{st}/3^{rd}$ quartiles; whiskers extend no further than 1.5 * IQR from the respective hinge. Source data for all panels are provided as a Source Data file.

---

resulting in diminished aECM invasion (Fig. 5), thus establishing a major pathogenetic role for Tks5-enabled podosomes and LF ECM invasion in pulmonary fibrosis. ECM invasion by non-leukocytes, a hallmark of cancer[12], is gaining increased attention in pulmonary fibrosis. IPF HLFs were shown to invade matrigel more efficiently than NHLFs or HLFs from other interstitial diseases[8,10,11]. Enhanced invasion correlated with increased actin stress fibers[10], and was suggested to be mediated, in part, by fibronectin and integrin α4β1 signaling[11], or hyalrounan (HA) and CD44 signaling[8]. CD44 has been localized in invadopodia in breast cancer cells and has been shown to be required for invadopodia activity[54], while TKS5+ LFs were found to preferentially express CD44 (Fig. 1f), suggesting that HA/CD44 participate in the regulation of podosome formation in LFs. TKS5+ LFs were also found to express preferentially CD274/PD-L1 (Fig. 1f), a proposed marker of invasive IPF LFs[55], suggesting yet another potential signaling input for podosome formation. Moreover, BALFs from BLM-treated mice or IPF patients stimulated ECM invasion of LFs[56,57], shown to be attenuated upon silencing LPAR1, EGFR and FGFR2 receptors[56], or by interfering with Sdc4-CXCL10 interactions[57], suggesting additional signals that could modulate LF invasion. HER2/EGFR2, a therapeutic target in breast cancer, has been also proposed to drive invasion in LFs[58], suggesting again similarities with metastatic ADC, and further suggesting repurposing a-HER-2 agents for the treatment of IPF. Interestingly, several of the identified invasion-associated genes (Fstl3, Il11, Hbegf, Ccn2, Inhba, Podxl, Sema7a, Bcl2a1b, Bcl2a1d, Sh3rf1) were found down regulated in the $Tks5^{+/-}$ invasion-defective LFs (Supplementary Data 1 and Supplementary Fig. 10d), further supporting the functional results on the role of TKS5 and podosomes in LF ECM invasion.

Given the expression of TKS5 in other cell types in IPF, like epithelial and basal cells (Fig. 1 and Supplementary Fig. 1) that are both intricately linked with IPF pathogenesis, a role for TKS5 in these cell types cannot be excluded. Podosome-like structures have been reported in epithelial cells, suggested to regulate their basement membrane adhesion[59,60], and thus, likely, re-epithelization, an essential process in wound healing. Moreover, pharmacologic inhibition of src-kinase, a master regulator of podosomes, was shown to attenuate IPF-basal cells-induced pulmonary fibrosis in minimally BLM-injured immunodeficient mice[61]. However, conditional epithelial deletion of Tks5 in future studies will be further required to dissect a possible pro-fibrotic role for TKS5 and podosomes in these cell types.

MMP9, a podosome enriched MMP (Fig. 2), has been previously found to be expressed in IPF, localized on reactive alveolar epithelial cells, basal-like cells, clusters of alveolar macrophages, as well as sub-epithelial fibroblasts[62,63]. Epithelial MMP9 has been suggested to partly mediate wound healing in keratinocytes via its proteolytic activity in podosomes, in association with CD44[60]. Moreover, CD44-bound MMP9 at the cell surface of cancer cells was shown to cleave latent TGFβ, and thus promote its activation[64], and thus, possibly, the activation of subepithelial fibroblasts in IPF. On the other hand, profibrotic Thy1- fibroblasts with increased migration potential were also shown to express MMP9 following stimulation with TGFβ[65]. As found here, post BLM LFs, that had increased Tks5/Col1a1 levels (Supplementary Fig. 7l, m) and increased podosomes (Supplementary Fig. 7o, p), also expressed higher MMP9 levels (Supplementary Fig. 7n). Although genetic deletion of MMP9 in mice (and/or MMP2) had minimal effects in Ad-TGFβ-induced pulmonary fibrosis[66], antibody-mediated MMP9 targeting demonstrated antifibrotic efficacy in a humanized immunodeficient model of IPF induced by IPF lung extracts, but only when fibrosis was promoted by "responder" IPF cells that had been shown to have reduced phosphorylated SMAD levels in response to a-MMP9 treatment in vitro[63]. Further studies will be required to appreciate the role of MMP9 in the pathogenesis of IPF and the possible therapeutic potential of targeting MMP9.

Connectivity MAP (CMap) analysis has emerged as an invaluable tool to connect gene expression, drugs and disease states[33]. CMap analysis of scRNAseq of IPF bronchial brushings suggested that src inhibition can reverse the observed pro-fibrotic transcriptional changes in IPF bronchial airway basal cells[61]. Moreover, CMap analysis of IPF transcriptional profiles and the nintedanib and pirfenidone corresponding transcriptional signatures, indicated src inhibition as the strongest connection[67]. As shown here, CMap analysis of the TGFβ-induced $Tks5^{+/-}$ LFs profile identified, among established and promising others, src inhibition as a possible treatment to limit LF invasion (Fig. 7) and therefore pulmonary fibrosis. In agreement, src inhibition was shown to reduce Tks5 levels and podosome numbers, to decrease aECM invasion and to reverse fibrosis in PCLS (Fig. 7) and to attenuate BLM-induced pulmonary fibrosis in vivo when administered by inhalation (Fig. 8). The aerosolized delivery, infrequent in IPF and animal models, opens new possibilities in IPF treatment, localizing treatment and avoiding systemic toxicity, as well as possibly increasing efficacy. The Src inhibitor Saracatinib, that has been previously also shown to attenuate BLM-induced pulmonary fibrosis[67], has recently entered clinical trials (NCT04598919), and if therapeutic effects are as efficient as in mice, the inhibition of podosome formation in LFs qualifies as a major part of its mode of action. Moreover, targeting kinase-mediated podosome formation, an inherent pathogenic LF property as shown here, as well as structural (TKS5 and its protein-protein interactions) or effector (MMP 2/9/14) podosome components, are very promising therapeutic targets in pulmonary fibrosis.

## Methods

All experimentation was performed according to the respective ethical regulations as outlined below.

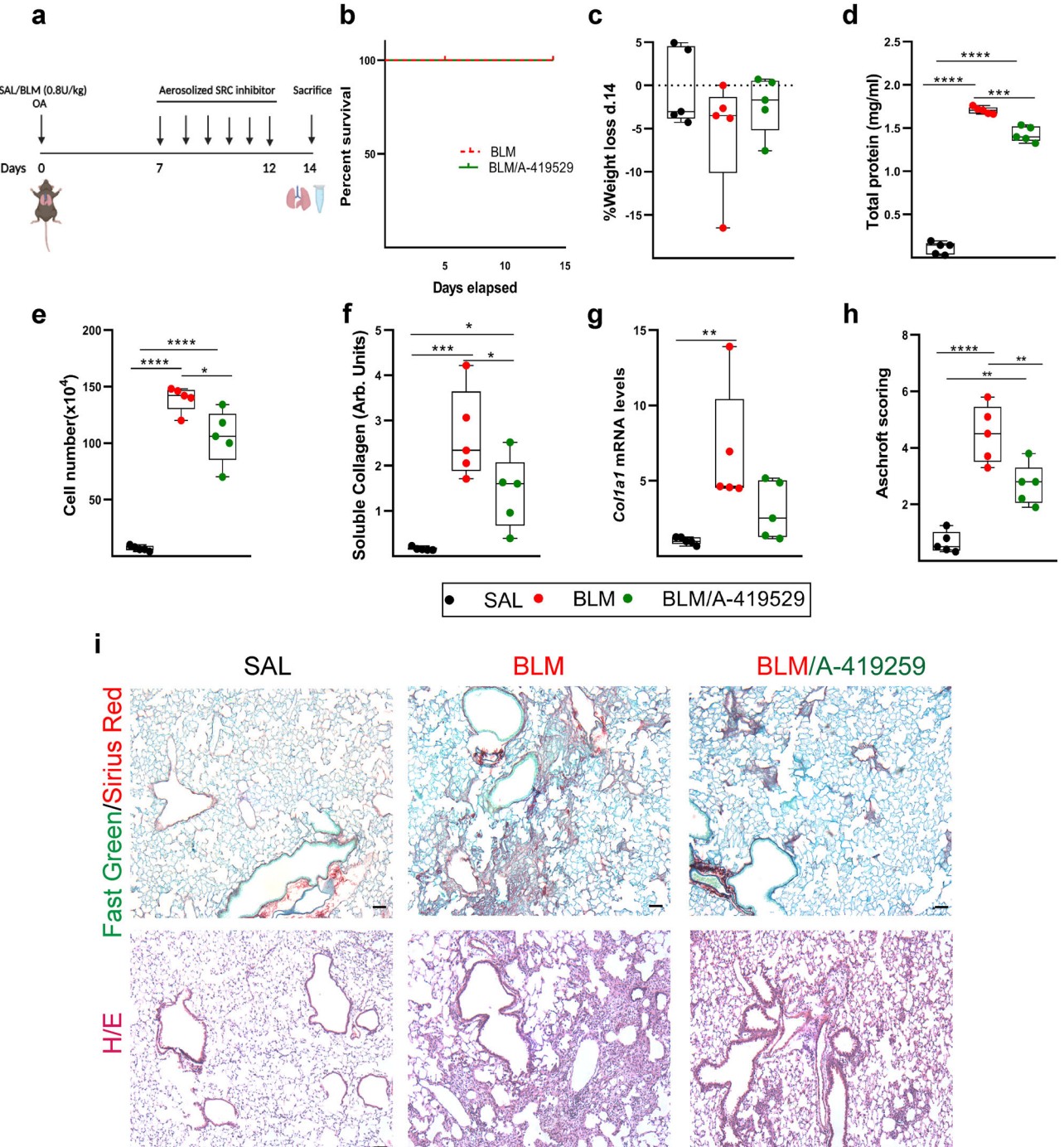

**Fig. 8 | Src inhibition attenuates bleomycin (BLM)-induced pulmonary fibrosis.**
**a** Schematic representation (biorender.com) of the BLM model and drug administration; $n = 5$. **b** Kaplan Meyer survival curve post BLM administration. **c** Weight change post BLM administration. **d** Total protein concentration in bronchoalveolar lavage fluids (BALFs), as determined with the Bradford assay. Statistical significance was assessed with two-tailed one-way ANOVA; $^{****}p < 0.0001$, $^{***}p = 0.0001$.
**e** Inflammatory cell numbers in BALFs, as counted with a hematocytometer. Statistical significance was assessed with two-tailed one-way ANOVA; $^{****}p < 0.0001$, $^{*}p = 0.0115$. **f** Soluble collagen levels in the BALFs were detected with the direct red assay. Statistical significance was assessed with two-tailed one-way ANOVA; $^{***}p = 0.0004$, $^{*}p = 0.0473$. **g** *Col1a1* mRNA expression was interrogated with Q-RT-

PCR. Values were normalized over the expression of the housekeeping gene *B2m* and presented as fold change over control. Statistical significance was assessed with two-tailed Kruskal Wallis; $^{**}p = 0.0094$. **h** Quantification of fibrosis severity in Hematoxylin & Eosin (H/E) stained lung sections via Ashcroft scoring. Statistical significance was assessed with two-tailed one-way ANOVA; $^{****}p < 0.0001$, $^{**}p = 0.0028/0.0074$. **i** Representative images from lung sections of murine lungs of the indicated genotypes, stained with Fast Green/Sirius Red (F.G/S.R; green/red) and H&E; scale bars = 50 μm. In all panels, all samples are biologically independent; boxplots visualize the median of each distribution; upper/lower hinges represent 1st/3rd quartiles; whiskers extend no further than 1.5 $^{*}$IQR from the respective hinge. Source data for all panels are provided as a Source Data file.

## Datasets

All analyzed, re-normalized datasets (Supplementary Table 1) were sourced from Fibromine;[21] Differentially expressed genes (DEGs): FC > 1.2, FDR corr. $p < 0.05$.

## Patients

All studies with human patient samples were performed in accordance with the Helsinki Declaration principles. Lung tissue samples (Supplementary Table 2) were obtained through the University of Pittsburgh Health Sciences Tissue Bank and Yale University Pathology Tissue service, a subset of previously well characterized and published samples;[68] studies had been approved by the Yale University Institutional Review Board (Yale IRB). LFs were isolated from the lung tissue of IPF patients and from the adjacent healthy tissue of patients undergoing open lung surgery for cancer (Supplementary Table 3) at the Department of Pulmonology, Bichat-Claude Bernard Hospital, Paris/France; studies were approved by the Committee for Personal Protection (CPP)−Ile de France 1 (#0911932). All patients consented in writing to the use of their samples for research purposes; no compensation was provided for their participation.

## Mice

Mice were bred at the animal facilities of Biomedical Sciences Research Center 'Alexander Fleming', under SPF conditions, at 20−22 °C, 55 ± 5% humidity, and a 12 h light-dark cycle; food (Mucedola diet #4RF21: humidity 12%, protein 18,5%, fat 3%, carbohydrate 53,5%, crude fibers 6%) and water were provided ad libitum. Mice were bred and maintained in their respective genetic backgrounds for more than 10 generations. All randomly assigned experimental groups consisted of littermate age-matched mice. The health status of the mice was monitored once per day; no unexpected deaths were observed. Euthanasia was humanly performed in a $CO_2$ chamber with gradual filling at predetermined time-points. All experimentation was approved by the Institutional Animal Ethical Committee (IAEC) of Biomedical Sciences Research Center "Alexander Fleming", as well as by the Veterinary Service of the governmental prefecture of Attica, Greece (# 8441/2017).

## Bleomycin-induced pulmonary fibrosis

Pulmonary fibrosis was induced by a single oropharyngeal administration (OA) of 0.8 U/kg bleomycin hydrogen chloride (BLM) (Nippon Kayaku Co., Ltd., Tokyo, Japan) at day 0 into anesthetized (i.p.; xylazine, ketamine, and atropine, 10, 100, and 0.05 mg/kg, respectively) 8-10-week-old male and female mice; control groups received sodium chloride (SAL). Dose and route were selected upon prior extensive local testing to induce a solid fibrotic profile, while minimizing lethality. All randomly assigned experimental groups consisted of littermate mice. Disease development was assessed in comparison with WT littermates 14 days post-BLM, at the peak of the disease (which resolves at d21 post BLM in these settings).

The A-419259 pharmacologic src inhibitor was administered directly in the mouse lungs through inhalation, using the inExpose system (Scireq, Cat.Number IX-XN1-T6) to conscious, softly re-strained mice. The inhibitor was administered at a therapeutic mode, once daily for 6 consecutive days, starting from day 7 post BLM administration. A-419529 was diluted in saline at a concentration of 0.182 mg/ml and 4 ml was administered for 5 mins in a group of 6 mice, corresponding to a final dose of 2 mg/kg per mouse. Control groups received aerosolized saline.

Following weighing, respiratory functions were measured with FlexiVent (SCIREQ, Montreal, Canada), according to manufacturer instructions and as previously described[24,43]. Briefly, a pressure-volume loop (PV) perturbation and a forced oscillation technique (FOT, single and low frequency) were applied in tracheotomized mice to produce the indicated measurements.

Bronchoalveolar Lavage fluid (BALF) was obtained by lavaging the lungs with 1 ml of 0.9% sterile sodium chloride three times. After the isolation, the samples were centrifuged at 1200 g for 10 min at 4 °C, the first BALF supernatant was stored at -80 °C for protein and collagen measurements. To estimate pulmonary inflammation, BALF cell pellets were redissolved in 1 ml saline, stained with 0.4% Trypan Blue solution and were counted with the use of a Neubauer hematocytometer. Total protein levels in BALFs, an indication of pulmonary edema and vascular leak, were assessed with the Bradford assay according to the manufacturer's instructions (Bio-Rad, Hercules, CA, USA). In a 96-well plate 5 µl of every BALF sample is placed, followed by the addition of 245 µl of 1x Bradford reagent (Serva/39222.03) and incubation for 5 minutes in the dark. Absorbance values were then measured at 595 nm, using a spectrometer, and were converted in concentration values (mg/ml) using a bovine serum albumin standard curve (BSA 0−2 mg/mL). Total soluble collagen in BALFs was quantified using the Sirius Red assay. 50 µl of BALF samples, diluted in 350 µl of 0.5 M acetic acid, were incubated for 30 min with 400 µl of Direct (365548-5 G Sigma-Aldrich) at RT, in the dark. This was followed by centrifugation, at 12000 g for 10 min and isolation of 200 µl of the supernatants. Absorbance values were measured at 540 nm, using a spectrometer, and were converted in concentration values (µg/ml) using a rat tail collagen I (C7661-5mg Sigma Aldrich) standard curve (0−500 µg/mL).

## Histology

The right lung was fixed overnight in 10% neutral buffered formalin and embedded in paraffin. 5µm lung sections were cut using a Microtome and stained with Hematoxylin/eosin (H&E) (Papanicolaou's solution HX16967353 Sigma Aldrich/ Eosin G CI45380 ROTH) with standard protocols. Fibrosis development was quantified by two independent reviewers, in a blinded manner, based on a modified Ashcroft score (0, normal lung; 1, isolated alveolar septa with gentle fibrotic changes; 2, fibrotic changes of alveolar septa with knot-like formation; 3, contiguous fibrotic walls of alveolar septa; 4, single fibrotic masses; 5, confluent fibrotic masses; 6, large contiguous fibrotic masses; 7, air bubbles; 8, fibrous obliteration). For Fast green-Sirius red (F.G/S.R) collagen staining, lung sections were deparaffinized in xylene and ethanol and incubated in Bouin's solution (75% picric acid/ 25% formaldehyde/ 1% acetic acid), for 1 hour at 56 °C, followed by staining with Fast Green (Glentham Life Sciences GT3407/100 g) 0,04% in picric acid for 15 minutes and Sirius Red 0,1%/Fast Green 0,04% dissolved in picric acid (197378-100 g Sigma-Aldrich) for 40 minutes. Stained sections were washed in acetic acid, then dehydrated and mounted with DPX (06522-500 ml Sigma-Aldrich). PCLS, isolated and cut as described below stained with H&E. For X-gal (Lac-Z), freshly isolated mouse lungs were inflated with 0,1 g/ml sucrose in 50% OCT/PBS, followed by the simultaneous embedding and freezing in OCT, using isopentane and dry ice. Sections of 6-10 µm were cut using a cryotome and fixed in 2% formaldehyde/ 0,2% glutaraldehyde for 15 minutes at 4 °C. Next, they were washed twice in cold PBS/ 2 mM $MgCl_2$ (PENTA) for 10 minutes and stained overnight with X-gal staining solution (2 mg/mL X-gal in 0.1 M Sodium phosphate buffer pH=7.3, 0,01% Sodium deoxycholate (30970 FLUKA), 5 mM $K_3Fe(CN)_6$ (60300 FLUKA), 5,7 mM $K_4Fe(CN)_6$ (12639 Riedel-de Haeen), 2 mM $MgCl_2$, 0,02% NP-40 (UN3082 Applichem) at 37 °C in the dark. The sections were then rinsed twice with PBS/2 mM $MgCl_2$ and $dH_2O$ for 5 minutes at room temperature, counterstained with eosin following by dehydration and mounting with DPX. Imaging was performed using a Nikon Eclipse E800 microscope (Nikon Corp., Shinagawa-ku, Japan) attached to a Q Imaging EXI Aqua digital camera, using the Q-Capture Pro 7 software. For immunohistochemistry studies, lung sections were deparaffinized in xylene, rehydrated in a gradient of ethanol, and briefly washed with water. The slides were kept in tap water until ready to

perform antigen retrieval with sodium citrate buffer with pH 6.0 by autoclave for 20 min. Then they were treated with blocking solution (10% normal goat serum/2% BSA; A9647 Sigma Aldrich) at room temperature for 1 h and incubated with primary antibodies overnight at 4 °C. After washing, they were incubated with fluorophore-conjugated secondary antibodies diluted in the blocking solution. Following this, sections were washed 3 times with PBS-T and mounted with medium containing DAPI (F6057, Sigma Aldrich) for nuclear visualization. Imaging was performed using a TCS SP8X White Light Laser confocal system (Leica).

### In vitro/ex vivo lung fibroblast cell model

Normal human lung fibroblasts (NHLFs) and IPF-HLFs were isolated from fresh tissue samples by plating several 2-3 mm pieces on 10 cm tissue culture plates in DMEM supplemented by penicillin/streptomycin solution, 10% FBS at 37° C and 5% $CO_2$ in a humidified atmosphere. 10-14 days following plating, proliferating fibroblasts surrounded the tissue pieces, which were then removed, and cells are detached with trypsin-EDTA solution and replated in F75 tissue culture flasks (P0) until confluent. Removed tissue pieces were replated in fresh 10 cm tissue culture plates for a second round of fibroblast outgrowth in the same conditions as above. Following 2-3 passages, homogeneous fibroblasts' colonies are observed that can be frozen down; upon thawing 3-4 passages are needed to obtain adequate numbers for experimentation; HLFs are used until passage 7-8. A similar procedure was independently used for an additional NHLF clone (Fig. S3A-B), as previously published[69].

Primary normal mouse lung fibroblasts (NMLFs) were isolated from 8-10-week- old C57Bl6/J mice and/or from BLM-challenged mice. Upon sacrifice, perfused lungs were excised in DMEM. Then lungs were minced and digested with 0.7 mg/ml collagenase type IV (C5138-Sigma), for 1 h at 37 °C. Digestion was followed by filtration and the suspension was centrifuged at 1200 g for 5 min. Finally, the pellet was resuspended in DMEM (GIBCO 41966-029) containing 10% FBS (GIBCO 10437-028/ origin Mexico. All experiments were performed at passage 2-3.

Murine and human embryonic LFs cell lines, 3T3 and MRC5 respectively, were purchased from ATCC (#CRL-1658 and #CCL-171 respectively).

NHLFs, IPF-HLFs, NMLFs, 3T3 and MRC5 cells were cultured in DMEM supplemented with 10% fetal bovine serum (FBS) and streptomycin/penicillin (GIBCO 15140-122) and amphotericin (GIBCO 15290-018) and incubated at 37 °C and 5% $CO_2$. Cells were cultured to 60-80% confluency, were starved overnight with serum-free DMEM (+ 0.1% BSA) and were exposed to 10 ng/ml recombinant human TGF-β1 (240-B-002 R&D SYSTEMS) for 24 h in serum-free DMEM. In control samples the dilutent of TGFβ (7.5% BSA in $H_2O$) was used. For pharmacologic studies, NHLFs were seeded at 6-well plates, were serum starved and pre-treated for 1 h with the indicated increasing concentrations of various agents and their diluents in controls. After one hour of pre-treatment, cells were incubated with TGFβ as usual.

The proliferation of all LF cultures was quantified with the MTT assay in 96-well plates, where a common solution of OPTI-MEM (GIBCO 11058-021) and MTT (0,7 mg/ml) (ACROS ORGANICS158990010) was added into each well. After incubation for 4 h and having confirmed the formation of purple crystals, the media was removed, and acidified isopropanol was added into each well to dissolve the formazan crystalline product. Absorbance values were determined at 570 nm and background subtracted at 660 nm using an OPTImax (Microplate Photometer (Molecular Devises).

The spreading and migration capabilities of LFs were assessed in a scratch wound assay. Cells were cultured into a 12-well plate as above and left to grow. Upon confluency, TGFβ was added, as described above. A wound was generated using a sterile pipette tip. The plate was placed in a tissue culture incubator at 37 °C, and photos were taken under a reverse microscope at specific time intervals.

Migration and invasion were quantified also with Boyden chambers (Costar 20122024/REF3422) according to manufacturer's instructions, as shown schematically in Fig. 5E. Briefly, LFs were added to the upper chamber which was pre-coated with aECM substrate (or not for migration) and allowed for 6 hours to invade/migrate through the transwell membrane to the lower side which was in touch with the starvation medium. Then, the cells which remained in the upper chamber were removed, while invasive or migratory cells, after washes and fixation, were stained with crystal violet. Additionally, stained cells were lysed with Lysis Buffer for 20 minutes and absorbance was measured at 550 nm using the TECAN Sunrise Microplate Photometer.

The proteolytic capacity of LFs was assessed with the Fluorescein gelatin degradation assay. 12-mm glass coverslips were acid washed with 20% nitric acid, incubated with 50 µg/ml poly-L-lysine, crosslinked with 0.5% glutaraldehyde and then coated for 20 min with 0.2% fluorescein-conjugated gelatin (Invitrogen, Gelatin from Pig Skin, Fluorescein Conjugate: G13187) in 2% sucrose (131621.1211Panreac) -containing phosphate buffered saline (PBS). Then, the coverslips were treated with sodium borohydride (NaBH4-Sigma 452882), washed with PBS, and transferred to a new 24-well-plate. Fibroblasts were seeded in gelatin-coated coverslips and treated with TGF-β1 as previously described. After 24 hours, medium was replaced with full medium and cells were processed for immunofluorescence, 24-48 hours later. The quantification of the degraded gelatin was analyzed using ImageJ, as previously described[70]. Briefly, the measurements of the degraded area, are reflected from the measurement of the "area fraction", which has been followed from the threshold adjustment that represents the actual degradation. Following this process, the "area fraction" values were then normalized to the number of nuclei as measured from the Dapi channel of the correlated image.

For immunofluorescence staining, cells were seeded in coverslips (20.000 cells/well) and after starvation, incubated with TGF-β1 as usual. The cells were fixed with 4% PFA for 15 min and permeabilized with 0.1% Triton X (T8532, Sigma Aldrich) for 10 min. This was followed by blocking with 2% BSA in PBS for 1 h at RT. The cells were incubated overnight at 4 °C with primary antibodies. The next day, cells were washed, and incubated with a secondary antibody and conjugated phalloidin in 1% BSA/PBS for 60 min at RT. Finally, after washing, coverslips were mounted with a drop of mounting Fluoroshield medium (containing DAPI for nucleus labeling). Confocal microscope images were analyzed with ImageJ. Signal colocalization was performed via the orthogonal views and k-curves analysis with ImageJ tools showing the intensity per z-stacking or per distance of the maximal projection accordingly.

The decellularization and generation of acellular ECM (aECM) from mouse lungs was performed based on similar protocols for other tissues[71,72]. Briefly, whole lungs were isolated and treated with increasing concentrations of SDS (Fisher Bioreagents BP166) (0.01, 0.1, 1%) in a PBS solution, with 24 h incubation for each SDS concentration. For the final step, decellularized lung tissues were washed with PBS for at least 3 days, cut into small pieces and stored at -80 °C. Frozen tissue was lyophilized using a lyophilizer and then milled in liquid nitrogen. To produce an ECM substrate, the milled form of the matrix was solubilized through enzymatic digestion. Pepsin (Sigma-Aldrich, P6887) was dissolved in 0.1 M HCl to make a concentration of 1 mg/ml. Approximately 10 mg of the ECM powder were digested in 1 mL of pepsin solution, in order to solubilize the ECM components. After approximately 48 hours, the matrix was diluted using 0.1 M acetic acid to make a 5 mg/ml concentration of lung ECM solution, which was used as a coating substrate for cells.

## Precision cut lung slices

C57-Bl/6, 8-10-week-old mice, were administered with SAL/BLM as described above. On day 11, mice were sacrificed and lungs, after perfusion, were inflated with 1 ml of with 1.5% Low Melting Agarose (15517-014-Invitrogen) in saline. Lungs were then isolated and incubated at RPMI medium (Thermo Fisher Scientific) supplemented with 10% FBS and 1% penicillin-streptomycin for 30 min at 4 °C to allow agarose polymerization. The left lobe of the agarose filled lungs, was cut into 200 μm slices (PCLS) with Vibratome. PCLS were then cultured in 700 μL of RPMI (GIBCO-21875-034) medium supplemented with 10% FBS and 1% penicillin-streptomycin in 24-well plates at standard conditions (37 °C and 5% $CO_2$) overnight. Next, PCLS were incubated to 2μm of A-419259 (Src-family inhibitor) and $H_2O$ for 3 consecutive days, changing the treatment daily. In day 14, PCLS were fixed overnight with PFA (Sigma-Aldrich P6148) at 4 °C, until slices embedded in paraffin. Finally, 5μm sections of PCLS were cut using a Microtome and stained for H&E, F.G/S.R and antibodies accordingly.

## Antibodies- reagents

Antibodies used in this study included anti-Tks5 (SH3 domain) rabbit monoclonal antibody (Merck, 3174822, 1:100), anti-SH3PXD2A mouse monoclonal antibody (Origene, clone OTI1F5-TA811757S, LOT F001, 1:250), col1a1 rabbit polyclonal antibody (Invitrogen, PA5-29569, LOT XK3738717 1:100), anti-A-actin (sma) mouse monoclonal antibody, (Origene, clone UM870129, LOT F001 1:250), recombinant Anti-Cortactin rabbit monoclonal (EP1922Y) antibody (Abcam, ab81208, 1:500), Alexa Fluor™ 633 Phalloidin (Invitrogen, A22284, LOT 2274768 1:50), MMP-9 XP Rabbit monoclonal (D6O3H) Antibody (Cell signaling #13667, LOT 3 1:100). Secondary antibodies included: Goat anti-Rabbit IgG (H + L) Cross-Adsorbed Secondary Antibody Alexa Fluor 488 (Life technologies, A11008, LOT1470706), Goat anti-Rabbit IgG (H + L) Cross-Adsorbed Secondary Antibody, Alexa Fluor 555 (Life technologies, A21428, LOT 1670185), Goat anti-Mouse IgG (H + L) Highly Cross-Adsorbed Secondary Antibody Alexa Fluor 488 (Life technologies, A11029, LOT1705900) Goat anti-Mouse IgG (H + L) Highly Cross-Adsorbed Secondary Antibody Alexa Fluor 555 (Life technologies, A22424, LOT1726548); all used in 1:500 dilution; all antibodies were selected from previous publication, while all antibodies were validated by the corresponding manufacturers for the specific employed methodologies (IHC/IF). For pharmacological studies A-419259 inhibitor SML0446 (Sigma-Aldrich), Nintedanib SML2848 (Sigma-Aldrich) and Pirferidone P2116 (Sigma-Aldrich) were used.

## Real Time quantitative RNA RT-PCR

In human samples, RNA was extracted from 30 to 50 mg of frozen lung tissue in 700 μL of Qiazol (Lysis buffer, Qiagen, Valencia, CA) by tissue disruption and homogenization using an electric homogenizer (Poly-Tron homogenizer H3660-2A, Cardinal Health, Dublin, OH) at 15.000 g for 15 seconds, according to the manufacturer's instructions. RNA was purified using the miRNeasy Mini kit (217004, Qiagen, Valencia, CA) with the assistance of the Qiacube automated system (9001292, Qiagen, Valencia, CA). The purity of the RNA was verified using NanoDrop at 260 nm and the quality of the RNA was assessed using the Agilent 2100 Bioanalyzer (Agilent, Technologies, Santa Clara, CA). Real-time PCR was performed with Taqman primers as described in the table below. Values were normalized to the expression of B2M.

In mouse samples RNA was extracted from the left lung lobe using the Tri Reagent (TR-118) obtained from Invitrogen and treated with DNAse (RQ1 RNAse-free DNAse) prior to RT-PCR according to manufacturer's instructions. cDNA synthesis was performed using 2 μg of total RNA per sample in 20-μl reaction using M-MLV RT (Promega). Real-time PCR was performed on a BioRad CFX96 Touch™ Real-Time PCR Detection System (Bio-Rad Laboratories). Values were normalized to the expression of b-2 microglobulin (b2m). The annealing temperature for all primers was 58 °C. Primers

sequences for RT and genomic PCRs are depicted at Supplementary Table 5.

## RNA sequencing

Six total RNA samples were prepared, and their concentration was measured with nanodrop (ND1000 Spectrophotometer−PEQLAB). The samples measured to a concentration of 400-500 ng/μl and therefore 1 μl of RNA, from each sample was used to proceed with the library preparation. The RNA quality of each sample was measured in bioanalyzer (Agilent Technologies) using the Agilent RNA 6000 Nano Kit reagents and protocol. For the preparation of per sample libraries, the 3′ mRNA-Seq Library Prep Kit Protocol for Ion Torrent (QuantSeq-LEXOGEN™ Vienna, Austria) was used according to manufacturer's instruction. Briefly, library generation was initiated by oligodT priming which contains the Ion Torrent compatible linker sequences. 5 to 500 ng per 5 μl of RNA from each sample was used to perform the first strand synthesis. After first strand synthesis any remaining RNA was removed and second strand synthesis was initiated by a random primer, containing Ion Torrent compatible linker sequences at its 5′ end, and a sequence polymerase. In line barcodes were introduced at this point. Second strand synthesis was followed by a magnetic bead-based purification step and the resulted purified library was amplified for 14 cycles and re-purified. Quality and quantity of each library was assessed in a bioanalyzer using the DNA High Sensitivity Kit reagents and protocol (Agilent Technologies). The quantified libraries were pooled together at a final concentration of 7 pM. The libraries pool was processed on the Ion Proton One Touch system where the libraries were templated and enriched using either the Ion PI™ Hi-Q™ OT2 200 Kit (ThermoFisher Scientific) and sequenced, with the Ion PI™ Hi-Q™ Sequencing 200 Kit on Ion Proton PI™ V2 chips (ThermoFisher Scientific) according to commercially available protocols. 3′ RNA-sequencing was performed on an Ion Proton™ System[73], according to the manufacturer's instructions. Initial analysis took place in Ion Torrent server.

## Single cell RNA-seq data re-analysis

Single cell RNA-seq data were downloaded from GSE122960 and processed with the R package Seurat (v.3.1.2 & 4.0.5)[74,75]. A similar to the original data analysis strategy was applied. Initially, each sample was processed on its own. After removing low-quality cells and genes, data were normalized using the LogNormalize method of Seurat and then top variable features were selected using the vst method. Data were scaled prior to principal component analysis (PCA) application and selection of the top principal components. The latter were used for a Shared Nearest Neighbor (SNN) graph-guided cell clustering and last, t-SNE dimensionality reduction was performed. Cell typing followed that of the original analysis as much as possible. Samples integration was performed with the standard Seurat v3 integration pipeline first for samples within each and then across phenotypes (donor and IPF). Integrated data were re-scaled, clustering and dimensionality reduction were repeated. Cell types inherited from single-sample analysis were validated and corrected whenever required according to the original publication of the dataset. Marker genes were identified using the Wilcoxon Rank Sum test applied on the "RNA" slot of the integrated samples object. Absolute fold change of at least 1.2 on natural scale and Bonferroni-corrected $p$-value < 0.05 were used as differential expression thresholds. Fibroblast sub-clusters were identified with a resolution of 0.1 after fibroblast cells were isolated from the rest of the dataset, re-scaled and new variable features were found as above.

## Data processing

Quant-Seq (Lexogen) FASTQ files obtained from the Ion Proton sequencing procedure were trimmed with Trim Galore (v.0.6.51) to remove low quality read ends using a Phred score of 20. Subsequently, a two steps alignment procedure was applied. Pre-processed reads

were aligned against the GRCm38 reference genome (Ensembl) with HISAT2 (v.2.2.1)[76] and then the reads left unmapped were subjected to a second alignment round using BOWTIE2 (v.2.3.5.1)[77] with the -local and -very-sensitive local switches turned on.

### Computational analysis

Downstream analysis of the resulting BAM files was performed with metaseqR2 (v.1.9.2)[78]. Briefly, the raw BAM files, one per sample, were summarized to a 3' UTR reads count table, using the package GenomicRanges (v1.44.0)[79] and Ensembl mouse genome mm10. For the UTR counting, the entire 3' UTR region, with a minimum length of 300 base pairs and 50 base pairs to flank the UTR end, was taken into consideration. In the resulting reads count table, each row represented one 3' UTR region and each column a Quant-Seq sample. Next, reads were summarized per gene and the returned gene count table was normalized using the package EDA-Seq (v.2.26.1)[80] after removing genes having zero reads across all samples. Post-normalization, gene counts were filtered for possible artifacts using default gene filtering options. The filtered gene counts table was subjected to differential expression analysis using sequentially all nine individual statistical analysis methods supported by metaseqR2. Their p-values were then combined by the PANDORA algorithm to account, among others, for the false positives reported. Benjamini-Hochberg corrected PANDORA p-values of less than 0.05 and absolute fold change of at least 1.2 were used as differential expression thresholds. Normalized expression values required for each heatmap were retrieved and standardized across samples. Hierarchical clustering of samples and genes based on calculated Euclidean distance.

### Gene set enrichment analysis

Differential expression analysis results were sorted by decreasing fold change and used for Gene Set Enrichment Analysis (GSEA) against Gene Ontology terms[81] using the package clusterProfiler (v.4.0.5)[82]. Signed normalized enrichment score (NES) was used to isolate the top of the significantly enriched induced (NES > 0) and suppressed (NES < 0) terms (adjusted p-value < 0.05).

### Text mining

PubMed 2022 baseline was downloaded from the respective FTP site. XML R package (v.3.99.0.8) was used to create an abstract-based corpus which was then queried with rentrez R package (v.1.2.3) for IPF[All Fields] OR (\"pulmonary fibrosis\"[MeSH Terms] OR \"pulmonary fibrosis\"[All Fields]) OR (\"lung diseases, interstitial\"[MeSH Terms] OR \"interstitial lung diseases\"[All Fields] OR \"interstitial lung disease\"[All Fields]) containing elements. Subsequently, human HGNC gene symbols atomization was performed using pubmed.mineR package[83] (v.1.0.19) and recovered genes were intersected with the human homologs of the mouse Quant-seq differentially expressed genes. Homolog feature mapping was obtained via biomaRt R package[84] (v. 2.48.3).

### Transcription factor analysis

Transcription factor analysis was performed using DoRothEA R package[85] (v.1.4.2). All mouse transcription factor-target regulons were queried for those high quality ones ("A" level of confidence) including as targets any of the Quant-seq differentially expressed genes. Subsequent filtering maintained those interaction pairs where both interactors were found significantly deregulated in the bulk sequencing experiment. Based on the mode of regulation (mor) as described in the DoRothEA database and the Quant-seq derived differential expression data, pairs with the same or opposite direction of deregulation were maintained in cases of an activator or repressor transcription factor, respectively.

### CMap/LINCS analysis

CMap/LINCS database (https://clue.io/query) query was performed using the top 150 up and top 150 down regulated genes as sorted by fold change. 150 consisted of the maximum number of genes supported by the platform. L1000 gene expression data from the Expanded CMap LINCS Resource 2020 (last update 11/23/2021) were queried. Results were processed by the R package cmapR (v.1.4.0). FDR-corrected p-values less than 0.05 were used to isolate those signatures having a statistically important connection with the provided one and signed normalized connectivity score (NCS) was used to discern between similar and opposite signatures. Focus was given on compounds (trt_cp) and peptides (trt_lig) having an already known mechanism of action and known targets. Datasets comprising of only one replicate and/or of a treatment duration other than 24 hours were discarded.

### Statistics

Statistical significance was assessed with the Prism (GraphPad) software according to its built-in recommendations, as detailed at each figure legend. Briefly, and unless otherwise stated, all datasets were tested for normal distributions via Shapiro-Wilk test, while all measurements were performed on distinct and independent samples. For the comparison of two normally distributed experimental groups, we employed the two-tailed unpaired t-test, for equal SDs or Welch's t-test for unequal ones. Not normally distributed data were analyzed with the two-sided Mann-Whitney test. Normally distributed multi parametric data with equal SDs were analyzed with the unpaired one-way ANOVA test and post hoc Tukey's test. Welch's ANOVA test coupled with post hoc Games-Howell test was used in cases of unequal SDs. For non-normally distributed multiparametric data, Kruskal-Wallis, followed by post hoc Dunn's test was utilized. For correlation analysis we used Pearson's, for normally distributed, or Spearman's correlation for non-normally distributed datasets. Most data are presented on box and whiskers graphs depicting the median as well as all experimental values (n).

### Image creation

Third party images were created at bioRender.com (under the relative agreements; 31/08/2023): Fig. 4a (DH25SQBJQT), Fig. 5e (XG25SQC37G), Fig. 8a (UC25SQ61BR), Sup Fig. 8a (VP25SQ8NTX), Sup Fig. 8b (GM25SQ96G1), Sup Fig. 8d (AV25SQA9RX), Sup Fig. 8f (ZI25SQ9VL2), and Sup Fig. 9a (BQ25SQ7828).

### Reporting summary

Further information on research design is available in the Nature Portfolio Reporting Summary linked to this article.

## Data availability

All re-analyzed publicly available datasets are listed in Supplementary Table 1, including accession numbers and hyperlinks. Quant-Seq data have been deposited at the GEO database under the accession code GSE220982. Already published, re-analyzed single cell RNA-seq data used are available at GEO database: GSE122960. All other relevant experimental data are within the paper and its supplementary information files. Source data are provided with this paper.

## Code availability

Data and code for the recreation of the computationally-created figures of the paper have been deposited at Zenodo (https://doi.org/10.5281/zenodo.8296510[86] and at https://github.com/dfanidis/TKS5_podosomes_IPF. Detailed scripts used are unrestrictedly available, within 10 days, upon request to Dionysios Fanidis (fanidis@fleming.gr; Institute for Fundamental Biomedical Research, Biomedical Sciences Research Center Alexander Fleming, Athens, Greece).

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

## Acknowledgements

We would like to thank Vassia Papadaki (Kafasla lab, BSRC Fleming) for support on confocal imaging, and Dimitris Kletsas (NCSR Demokritos) for NHLF clones and advice on optimal NHLF culture conditions.

This research was co-financed by Greece and the European Union (European Social Fund—ESF) through the Operational Program «Human Resources Development, Education and Lifelong Learning» in the context of the project "Strengthening Human Resources Research Potential via Doctorate Research" (MIS-5000432), implemented by the State Scholarships Foundation (Fellowship to IB). Research was further partly supported through the Hellenic Foundation for Research and Innovation (HFRI) under the "2nd Call for HFRI Research Projects to support Faculty Members & Researchers" (#3565 to VA). Human studies were supported by the NIH NHLBI grants R01HL127349, R01HL141852, U01HL145567, and UH2HL123886 to NK. The funders had no role in the design of the study, or in the collection, analyses, or

interpretation of data, or in the writing of the manuscript, or in the decision to publish the results.

## Author contributions

I.B. and P.K. performed most presented experiments and analyzed the relative data, assisted by E.-D.N., D.N., A.G., and E.T.; A.T. performed human studies supervised by R.H., N.K. and V.A.; D.F. performed in silico data re-analysis and supervised all statistical analyses. V.H. and P.H. performed RNAseq. K.A., B.C. and N.K. provided resources. V.A. conceived and coordinated the project. I.B., P.K., D.F. and V.A. wrote the paper, which was edited by V.A. and critically commented by all co-authors.

## Competing interests

A.T. has received fees for speaking and/or organizing education from AstraZeneca, Menarini, Boehringer Ingelheim, Chiesi, Hoffmann-La Roche, Ltd., GlaxoSmithKline and Elpen, for consulting from Boehringer Ingelheim, Pfizer, Gilead, Hoffmann-La Roche, Ltd., GlaxoSmithKline, and has received research funding, including institutional funding, from Boehringer Ingelheim, Chiesi, Hoffmann-La Roche, Ltd., GlaxoSmithKline and Astra Zeneca, outside the submitted work. B.C. has received fees for speaking and/or organizing education from Apellis, Astra Zeneca, BMS, Boehringer Ingelheim, Novartis, Roche and Sanofi, for consulting fees from Apellis, BMS, Boehringer Ingelheim and Sanofi, and has received research funding from Boehringer Ingelheim, outside the submitted work. N.K. is a scientific founder at Thyron, served as a consultant to Biogen Idec, Boehringer Ingelheim, Third Rock, Pliant, Samumed, NuMedii, Theravance, LifeMax, Three Lake Partners, Optikira, Astra Zeneca, RohBar, Veracyte, Augmanity, CSL Behring, Galapagos and Thyron over the last 3 years, reports Equity in Pliant and Thyron, and grants from Veracyte, Boehringer Ingelheim, BMS and non-financial support from MiRagen and Astra Zeneca, outside the submitted work. Other authors declare that they have no conflict of interest.
