## [Peer Review File · Nature Communications]

REVIEWER COMMENTS

Reviewer #1 (Remarks to the Author):

Barbayianni et al. have studied the effect of TKS5 on the development of fibrotic lung disease. Their findings support a role for TKS5 enabling podosome formation and potentially involved in the recruitment and activation of lung fibroblasts in the fibroproliferative process. Although the authors present interesting in vitro, experimental murine, and human findings, the mechanisms underlying these observations are not fully described and additional experiments need to be included to understand the temporal/causal relationship between TKS5 and the other markers of lung fibrosis. More specifically, the following topics would benefit from further development.

1. Key question is whether TKS5 expression in lung fibrosis is involved in the mechanism of lung fibrosis or the result of lung fibrosis. Since TKS5 has been observed in embryonic development, this remains a critical question that is relevant to cause/effect of lung fibrosis. Moreover, the investigators have shown that TGF- β , as well as a stiff Col1a1-rich acellular fibrotic ECM induce Tks5 expression. So then is TKS5 the cause or result of lung fibrosis?
2. The ability of human lung fibroblasts from IPF patients in comparison to controls to produce podosomes in vitro is quite interesting, however, it appears that this required 7-8 passages of lung fibroblasts. This finding would be more robust if earlier passage lung fibroblasts were studied.
3. The cell specificity of TKS5 needs further attention. The investigators have focused on the expression of TKS5 in lung fibroblasts, however, among those with IPF, TKS5 is also overexpressed in epithelial cells and basal cells. What is the role of TKS5 in these cells?
4. The murine model of Tks5 insufficiency (Tks5 haploinsufficient mice) demonstrated reduced lung fibrosis in comparison to WT mice when challenged with bleomycin. However, these studies were terminated 2 weeks after challenge with bleomycin. This is a rather short exposure period to bleomycin and the authors should either explain why they used this shorter timeline or redo the experiments and harvest the mice 3 weeks after challenge with bleomycin. Also, it's unclear whether the Ashcroft scores were performed with PCLS or post in vivo bleomycin challenges.
5. Src-inhibition studies should be performed in vivo with WT and Tks5 haploinsufficient mice, examining the expression of Tks5, Col1a1, and quantifying fibrosis.

6. UIP involves more than just fibroblastic foci pathologically. These additional histo-pathological features should be discussed in the introduction.

Reviewer #2 (Remarks to the Author):

Idiopathic pulmonary fibrosis (IPF) is a widespread disease with unknown etiology. Current treatments are not curative and provide only minor improvement. Targeting of Src signalling in IPF is being evaluated clinically, and this signaling pathway has shown significant promise for enhanced treatment options. In this work, Barbayianni et al. show that Src/TKS5 drives podosome formation which promotes invasive behaviour in IPF fibroblasts. Further targeting this pathway in a mouse model reverses the progression of pulmonary fibrosis. Overall, the manuscript is well written, provides significant novel insight, and conducts the necessary experimental controls to support the work.

Before being suitable for publication, however, a number of changes/revisions are requested.

Comments:

1. Pages and page lines should be numbered to make reviewing easier.
2. Fig. 1G: x-axis is not labeled
3. Fig. 1I: Authors claim that Tks5 staining (green) is largely overlapping with Coll1a1 or α -SMA staining (red) however representative images show little (Tks5/Coll1a1) or no colocalization. The vast majority of the cells present only one type of staining. Better images that support the authors' claim must be provided, ideally with co-localization calculated using one of the standard methods. The figure would benefit from showing a phase contrast panel for (I).
4. Fig. S3 E-G: MTT and t-scratch assay results appear to be not mentioned in the manuscript text.
5. Fig. S4 D-E seems to be mislabeled (i.e., E should be intensity curves, D includes higher mags of Cs).
6. Fig. 2E, G: Unlike CTTN that appears to strongly localize to podosomes in response to TGF stimulation (Fig. 2E), the majority of MMP localized in Golgi with or without TGF (Fig. 2E). Co-localization of CTTN with MMP9 must be shown to support author's claim of TGF-induced localization of MMP9 to podosomes.
7. Fig. 2K: (FITC degradation) does not appear to correlate to the quantification plot showing 25-35% of FITC degradation. The degradation area shown in the representative image appears to be substantially below the minimal value of ~25%. From the "Methods" section it is not clear if the degradation area was normalized to the cell number or not. Please clarify. Cell density appears to be different in Fig. 2H, K.

8. Fig. S5: IPF 799 and IPF 812 cells appear to be at much higher density than the control. Was the degradation area normalized to cell number? Figures would benefit from showing the DAPI-only channel.

9. Fig. S6 seems to be mislabeled in the manuscript text: “Moreover, and as in the case of IPF LFs, mouse primary LFs isolated post-BLM administration presented with prominent podosome rosettes in the absence of any stimulation (Fig. S6L-N)”. It appears that the correct labelling should be Fig. S6N-O.

10. Discussion section is mainly a recapitulation of the Results – an opportunity is missed to contextualize the significance of the work, ie discuss how the work relates to other therapeutically targetable mechanisms that drive fibroblast invasion in IPF (i.e. MMP function).

Response to reviewer's comments

We would like to thank both reviewers as their comments have significantly improved our manuscript. Please find below our point-by-point responses.

Overall, we have added a new figure with the requested *in vivo* treatment, and a new supplementary figure with imaging controls, we split figure 2 to two figures, and we added several panels in existing figures.

We also significantly enriched the discussion part, as prompted by the reviewers, to whom we are again grateful.

Please note, that due to length restrictions the title and the abstract have been shortened. Moreover, additional statistical information was added to the figure legends, per journal mandate.

We hope that the reviewers will now find the manuscript suitable for publication.

Reviewer #1 (Remarks to the Author):

*Barbayaanni et al. have studied the effect of TKS5 on the development of fibrotic lung disease. Their findings support a role for TKS5 enabling podosome formation and potentially involved in the recruitment and activation of lung fibroblasts in the fibroproliferative process. Although the authors present interesting *in vitro*, experimental murine, and human findings, the mechanisms underlying these observations are not fully described and additional experiments need to be included to understand the temporal/causal relationship between TKS5 and the other markers of lung fibrosis. More specifically, the following topics would benefit from further development.*

1. Key question is whether TKS5 expression in lung fibrosis is involved in the mechanism of lung fibrosis or the result of lung fibrosis. Since TKS5 has been observed in embryonic development, this remains a critical question that is relevant to cause/effect of lung fibrosis. Moreover, the investigators have shown that TGF- β , as well as a stiff Colla1-rich acellular fibrotic ECM induce Tks5 expression. So then is TKS5 the cause or result of lung fibrosis?

Of course, this is an important question that applies in many necessary developmental programs/pathways that are aberrantly reactivated in IPF pathogenesis, e.g. wnt signaling, and ATX/LPA signaling. While the answer is difficult to obtain experimentally for all such pathways, regarding TKS5 and podosomes in particular, since they are induced by pro-fibrotic factors, we believe, as we have stated, that “*podosome formation is an unappreciated central response of LFs to pro-fibrotic factors*”. As also stated, “*the formation of podosomes in LFs upon mechanical cues from the stiff ECM of fibrotic lungs is a major component of the suggested crosstalk of ECM with fibroblasts^{42,43}, especially considering the age-related increase of ECM stiffness in the lungs⁷, and the suggested role of mechanosensitive signaling in LF activation and pulmonary fibrosis⁴⁴.*” Moreover, and as we have shown, “*the stiff fibrotic post-BLM aECM was shown to stimulate Tks5 expression and podosome formation in LFs and to perpetuate the increased expression of Colla1 (Fig. 5)*”.

Taken together, we believe that the formation of podosomes is a developmental program that gets aberrantly re-activated upon IPF from pro-fibrotic factors, and that perpetuate LF activation and stimulate, at least in part, ECM invasion and LF accumulation and thus pulmonary fibrosis. A relative paragraph was added in the discussion. We are grateful to the reviewer for his comment.

2. The ability of human lung fibroblasts from IPF patients in comparison to controls to produce podosomes in vitro is quite interested, however, it appears that this required 7-8 passages of lung fibroblasts. This finding would be more robust if earlier passage lung fibroblasts were studied.

In our opinion, the finding is more robust as the number of passages increase and not the opposite. The longer the fibroblasts retain their podosomes, the stronger the statement “*podosome formation is an inherent property of IPF lung fibroblasts*” becomes. Regardless, and in practical terms, it is impossible for us to have adequate numbers with fewer passages (<7-8). It takes 2-3 passages to freeze down the initial cultures (the same is true for ATCC procured IPF LFs), performed in the lab of B. Crestani, and another 3-4 passages for the included experimentations (performed in the Antoniou and Aidinis labs). Additionally, a new publication was published 3 days after the finalization of our manuscript showing invadosomes at earlier passages (3-7). Moreover, the podosomes in mouse LFs were detected at passage 3, where we perform all *ex vivo* studies with mouse LFs (Fig. S5). We also have to note that podosomes could be seen in IPF LFs or TGF-induced wt LFs long time ago, but were not identified, commented upon or further researched. E.g. : 29046395 (Fig. 3), 26399448 (Fig. 3), 30166321 (Fig. 6).

To accommodate the reviewer’s concern, the said publication is now included and discussed, while the NHLF M&Ms (online sup) has been updated to include a relative statement on LF passaging.

3. The cell specificity of TKS5 needs further attention. The investigators have focused on the expression of TKS5 in lung fibroblasts, however, among those with IPF, TKS5 is also overexpressed in epithelial cells and basal cells. What is the role of TKS5 in these cells?

Indeed TKS5/Tks5 expression could be detected in epithelial and basal cells, while both cell types are decisively linked with IPF pathogenesis. However, there is limited available information on a possible role for Tks5 or podosomes in these cells. Most noteworthy, “pharmacologic inhibition of src-kinase, a master regulator of podosomes, was shown to attenuate IPF-basal cells-induced pulmonary fibrosis in minimally BLM-induced injury immunodeficient mice”. One new paragraph was added in the discussion. As we state there: “*conditional epithelial deletion of Tks5 in future studies will be further required to dissect a possible pro-fibrotic role for TKS5 and podosomes in these cell types*”.

4. *The murine model of Tks5 insufficiency (Tks5 haploinsufficient mice) demonstrated reduced lung fibrosis in comparison to WT mice when challenged with bleomycin. However, these studies were terminated 2 weeks after challenge with bleomycin. This is a rather short exposure period to bleomycin and the authors should either explain why they used this shorter timeline or redo the experiments and harvest the mice 3 weeks after challenge with bleomycin.*

Also, it's unclear whether the Ashcroft scores were performed with PCLS or post in vivo bleomycin challenges.

As stated in the online supplement, “*Disease development was assessed in comparison with WT littermates 14 days post-BLM, at the peak of the disease (which resolves at d21 post BLM in these settings).*” Moreover, as we also state in the online supplement, “*Dose and route were selected upon prior extensive local testing to induce a solid fibrotic profile, while minimizing lethality*”. All these are also included and discussed in our “BLM-revisited” publication (PMID: 30320115), an experimental follow up of our previous reviews on the BLM model (PMID: 21832918 and 28804709). As discussed there, “*Beyond the route of administration, the severity of BLM effects highly depends on the precise genetic background of mice (i.e. C57Bl6 J vs N, further differing between vendors), the local genetic drift of the colony and the health status of the corresponding animal house*”. Therefore, timing of sacrifices was optimal.

However, to accommodate the reviewer’s concern and to avoid possible confusion of readers, one phrase was added in the corresponding results’ paragraph, stating that experimental mice “... *were sacrificed 14 days post BLM at the peak of the disease in these settings, ...*”.

The legend of the corresponding figure 3 has been amended to indicate that Ashcroft scoring was performed in H/E-stained lung sections.

5. *Src-inhibition studies should be performed in vivo with WT and Tks5 haploinsufficient mice, examining the expression of Tks5, Col1a1, and quantifying fibrosis.*

Following the reviewer’s recommendation, we administered the same src-inhibitor used *ex vivo* to mice undergoing BLM-induced pulmonary fibrosis, in a therapeutic mode, 7d post BLM (disease peak 14d). More importantly, the inhibitor was administered aerosolized, thus avoiding possible systemic toxicity, and possibly enhancing its local efficacy. The results, now included as figure 8, show a clear protection from disease development and further support src and podosomes as therapeutic targets in IPF.

6. *UIP involves more than just fibroblastic foci pathologically. These additional histopathological features should be discussed in the introduction.*

We apologize for the oversimplification of the UIP profile emphasizing fibroblasts. The corresponding part has been updated according to the already cited guidelines paper.

Reviewer #2 (Remarks to the Author):

Idiopathic pulmonary fibrosis (IPF) is a widespread disease with unknown etiology. Current treatments are not curative and provide only minor improvement. Targeting of Src signalling in IPF is being evaluated clinically, and this signaling pathway has shown significant promise for enhanced treatment options. In this work, Barbayianni et al. show that Src/TKS5 drives podosome formation which promotes invasive behaviour in IPF fibroblasts. Further targeting this pathway in a mouse model reverses the progression of pulmonary fibrosis. Overall, the manuscript is well written, provides significant novel insight, and conducts the necessary experimental controls to support the work.

Before being suitable for publication, however, a number of changes/revisions are requested.

Comments:

1. Pages and page lines should be numbered to make reviewing easier.

Page and line numbering has been added accordingly as requested.

2. Fig. 1G: x-axis is not labeled

The axis labeling has been added, as requested.

3. Fig. 1I: Authors claim that Tks5 staining (green) is largely overlapping with Coll1a1 or a-SMA staining (red) however representative images show little (Tks5/Coll1a1) or no colocalization. The vast majority of the cells present only one type of staining. Better images that support the authors' claim must be provided, ideally with co-localization calculated using one of the standard methods. The figure would benefit from showing a phase contrast panel for (I).

Indeed, the overlap of Tks5 and Coll1a1 staining that was detected was confined to 20% as shown in the accompanying graph, and as opposed to aSMA/2%, a percentage now included in the text for emphasis. This is further supported by the correlation of Tks5 and coll1a1 observed though out in different experimental settings.

As requested, we have retaken representative images and performed 3D surface analysis that clearly supports Tks5 and coll1a1 co-expression in an LF subset. A new supplementary figure was created showing the 3D surface analysis, and separately the different channels as requested.

4. Fig. S3 E-G: MTT and t-scratch assay results appear to be not mentioned in the manuscript text.

These well-known TGF effects have now been added to the text, as requested.

5. Fig. S4 D-E seems to be mislabeled (i.e., E should be intensity curves, D includes higher mags of Cs).

The higher magnifications (D) were removed, and the area for K-curve analysis are now indicated in (C).

6. Fig. 2E, G: Unlike CTTN that appears to strongly localize to podosomes in response to TGF stimulation (Fig. 2E), the majority of MMP localized in Golgi with or without TGF (Fig. 2E). Co-localization of CTTN with MMP9 must be shown to support author's claim of TGF-induced localization of MMP9 to podosomes.

We have now added a new double staining image showing TKS5 (instead of CTTN) and MMP9 colocalization in podosomes, as well as the corresponding orthogonal and K-curve analysis (In Fig. S5G-H). We also have to note that little is known on MMP9 spatiotemporal regulation within cells, as well as on its expression from other cell types, e.g. macrophages. We have tried in this manuscript not to delve much into MMP9, the focus of a follow up.

7. Fig. 2K: (FITC degradation) does not appear to correlate to the quantification plot showing 25-35% of FITC degradation. The degradation area shown in the representative image appears to be substantially below the minimal value of ~25%. From the "Methods" section it is not clear if the degradation area was normalized to the cell number or not. Please clarify. Cell density appears to be different in Fig. 2H, K.

It should be noted that the 25% degradation of IPF LFs should be compared with the 10% of wt LFs (in an almost green image). Nevertheless, as the reviewer very correctly indicated, we have now also included a graph where gelatin degradation area was normalized to cell numbers.

Concerning cell density, in all experiments we seeded identical numbers from the different clones, which are not proliferating with equal efficiencies. This is the reason why we have performed most experiments at different densities (Fig. S6A), to exclude similar concerns. A phrase was added in the results to point to the sup density data.

8. Fig. S5: IPF 799 and IPF 812 cells appear to be at much higher density than the control. Was the degradation area normalized to cell number? Figures would benefit from showing the DAPI-only channel.

Please see our response to the similar comment above.

The DAPI-only channel is now showing at Sup Fig. 6C.

9. Fig. S6 seems to be mislabeled in the manuscript text: "Moreover, and as in the case of IPF LFs, mouse primary LFs isolated post-BLM administration presented with prominent podosome rosettes in the absence of any stimulation (Fig. S6L-N)". It appears that the correct labelling should be Fig. S6N-O.

Labeling has been corrected.

10. Discussion section is mainly a recapitulation of the Results – an opportunity is missed to contextualize the significance of the work, ie discuss how the work relates to other therapeutically targetable mechanisms that drive fibroblast invasion in IPF (i.e. MMP function).

We have tried to keep the necessary comparative summarization of results to a minimum of ~20 lines. We have now added several new paragraphs to the discussion, as prompted by the reviewers above, including one focusing on MMP9 and the therapeutic potential of its targeting. To further support the new discussion paragraph on MMP9, we have added some introductory info and some additional experimental data on MMP9. We are grateful for the comment.

REVIEWERS' COMMENTS

Reviewer #1 (Remarks to the Author):

The authors have sufficiently addressed the concerns raised, and have modified the data presented and the text accordingly. The presentation is clear and the conclusions are consistent with the findings. Their observations are important to the field of fibrotic lung disease, advancing concepts about the pathogenesis and potential therapeutic targets.

Reviewer #2 (Remarks to the Author):

The authors sufficiently addressed our concerns and I can now recommend the manuscript for publication

Response to reviewer's comments

We would like to thank both reviewers as their comments have significantly improved our manuscript.

Reviewer #1:

The authors have sufficiently addressed the concerns raised, and have modified the data presented and the text accordingly. The presentation is clear and the conclusions are consistent with the findings. Their observations are important to the field of fibrotic lung disease, advancing concepts about the pathogenesis and potential therapeutic targets.

Reviewer #2:

The authors sufficiently addressed our concerns, and I can now recommend the manuscript for publication